# AOR: Anatomical Ontology-Guided Reasoning for Medical Large Multimodal Model in Chest X-Ray Interpretation

**Qingqiu Li**[1], **Zihang Cui**[2], **Seongsu Bae**[3], **Jilan Xu**[1], **Runtian Yuan**[1], **Yuejie Zhang**[1,✉],
**Rui Feng**[1,✉], **Quanli Shen**[4], **Xiaobo Zhang**[4,✉], **Shang Gao**[5], **Junjun He**[6], **Shujun Wang**[7,8]

[1] College of Computer Science and Artificial Intelligence,
Shanghai Key Laboratory of Intelligent Information Processing, Fudan University
[2] School of Computer Science and Technology, Xidian University
[3] Kim Jaechul Graduate School of AI, Korea Advanced Institute of Science and Technology
[4] National Children's Medical Center, Children's Hospital of Fudan University
[5] School of Information Technology, Deakin University
[6] Shanghai Artificial Intelligence Laboratory
[7] Department of Biomedical Engineering, The Hong Kong Polytechnic University
[8] Research Institute for Smart Ageing, The Hong Kong Polytechnic University
{yjzhang,fengrui}@fudan.edu.cn, zhangxiaobo0307@163.com

## Abstract

Chest X-rays (CXRs) are the most frequently performed imaging examinations in clinical settings. Recent advancements in Medical Large Multimodal Models (MLMMs) have enabled automated CXR interpretation, improving diagnostic accuracy and efficiency. However, despite their strong visual understanding, current MLMMs still face two major challenges: (1) insufficient region-level understanding and interaction, and (2) limited accuracy and interpretability due to single-step prediction. In this paper, we address these challenges by empowering MLMMs with anatomy-centric reasoning capabilities to enhance their interactivity and explainability. Specifically, we propose an **A**natomical **O**ntology-Guided **R**easoning (AOR) framework that accommodates both textual and optional visual prompts, centered on region-level information to enable multimodal multi-step reasoning. We also develop AOR-Instruction, a large instruction dataset for MLMMs training, under the guidance of expert physicians. Our experiments demonstrate AOR's superior performance in both Visual Question Answering (VQA) and report generation tasks. Code and data are available at: `https://github.com/Liqq1/AOR`.

## 1 Introduction

Chest X-rays (CXRs), with over 2 billion examinations annually [2], are essential for diagnosing and monitoring thoracic diseases [34]. However, their interpretation is time-consuming and expertise-dependent, leading to diagnostic delays, variability, and errors [5, 6], especially under increasing clinical workloads and radiologist shortages. These challenges highlight the urgent need for accurate and efficient automatic CXR interpretation systems to support modern medical practice.

Early task-specific models for CXR interpretation focused on disease classification [32], detection [28], or report generation [8], but often lacked generalization and interpretability. Recently, the advancement of Large Multimodal Models (LMMs) [23, 22] has emerged as a promising and scalable solution for CXR interpretation. Notable examples include LLaVA-Med [18] and CheXagent [9].

---

✉ Corresponding author

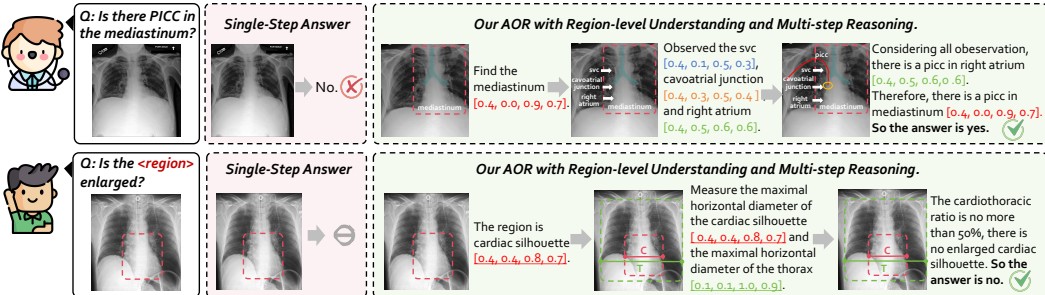

Figure 1: Previous image-level MLMMs (shown in light red) make incorrect predictions or fail to predict due to (1) insufficient region-level perception and (2) reliance on single-step reasoning. In contrast, our AOR model (shown in light green) delivers explainable and accurate answers by (1) emphasizing region-level understanding and (2) employing multi-step reasoning.

While these existing Medical LMMs (MLMMs) have shown remarkable capabilities in both visual understanding and language modeling, they still face two key challenges that significantly limit their effectiveness in medical imaging applications:

**(1) Insufficient Region-Level Understanding and Interaction** Radiologists interpret CXRs by scanning the full image and then examining specific anatomical regions for abnormalities. To emulate this process, models must capture fine-grained visual details, spatial relationships, and anatomical hierarchies. However, current image-level MLMMs [18, 36] struggle to detect subtle but clinically relevant findings. Additionally, as shown in Fig. 1, region-level interaction is crucial: not only for radiologists revisiting suspicious areas but also for non-experts who rely on models to interpret visual cues without medical terminology. Existing MLMMs lack such capabilities, limiting their real-world applicability. Thus, it is imperative to highlight region-level perception to overcome these obstacles.

**(2) Limited Accuracy and Interpretability due to Single-Step Prediction Without Reasoning** Medical imaging often involves overlapping symptoms and diverse disease manifestations, demanding multi-step reasoning for accurate diagnosis. Models must jointly consider lesion location, appearance, and clinical context in relation to the question. However, current MLMMs typically make single-step predictions without explicit reasoning, resulting in misinterpretations [10], hallucinations [29], and poor symptom-lesion-disease alignment. A key bottleneck is the lack of high-quality, clinically grounded instruction data to support multi-step reasoning, given the specialized knowledge and low error tolerance required. To address this, constructing a clinically credible Chain-of-Thought (CoT) dataset is essential for improving both accuracy and interpretability in MLMMs.

In this paper, we introduce the **Anatomical Ontology-Guided Reasoning** (AOR) framework with a three-stage training strategy. AOR centers on the anatomical regions relevant to the given question, incorporating their positional and representational information to enable multimodal, multi-step reasoning. Then, to address the shortage of high-quality multimodal reasoning datasets for MLMMs, we develop the AOR-Instruction dataset under the guidance of three expert physicians. This dataset consists of two subsets: **AOR-VQA** and **AOR-RG**. Specifically, for AOR-VQA, we construct 2,812 CoT templates based on three anatomical ontologies to provide precise CoT answers for 290k Visual Question Answering (VQA) samples. For AOR-RG, 133k CXR-report pairs are used for full image report generation, and raw reports are further decomposed into fine-grained descriptions, yielding 399k strictly aligned region–sentence pairs for interpretable region report generation.

By empowering Medical LMMs with anatomy-centric reasoning capabilities, we introduce a new paradigm toward interactive and explainable medical imaging analysis for CXR interpretation. The contributions of this work are summarized as follows.

- We propose an Anatomical Ontology-Guided Reasoning (AOR) framework, which supports both textual and optional visual prompts (i.e., region-of-interest (ROI) crops) as input, centered on region-level information to enable multimodal, multi-step reasoning.

- We develop a large instruction dataset named AOR-Instruction by leveraging anatomical regions and their ontologies. It consists of two parts: AOR-VQA for VQA and AOR-RG for full image and region report generation, containing 290k and 532k data pairs, respectively.

- Extensive experiments demonstrate the superiority of AOR, which outperforms the second-best MLMM by an average of 6.81% on the VQA and 5.27% on report generation, underscoring the crucial role of region perception and reasoning capabilities in supporting clinical decision-making.

## 2 Related Works

**Medical Large Multimodal Models** With the success of LLMs [1, 37, 38], researchers are enhancing these models by incorporating visual understanding capabilities, leading to the emergence of LMMs [23, 22]. In the medical domain, numerous LMM-based studies have also arisen. Prominent models such as LLaVA-Med [18] and Med-Flamingo [27] first perform image-text feature alignment using paired medical data, followed by meticulously designed instruction tuning. Although these models exhibit strong visual understanding, they are primarily limited to image-level tasks like report generation and medical visual question answering. They do not explicitly learn region-level features during the training process, which constrains their region-level perception.

**Region-Level Medical LMMs** To achieve more fine-grained image understanding, recent research has integrated region-level data into the training of LMMs. Shikra [7] directly quantizes bounding boxes into coordinates (numerical representations of positions). Subsequently, GPT4RoI [42] and RegionGPT [11] extract region features from the original images and include them as part of the input token sequences, allowing the models to comprehend region representations and enabling them to process visual prompts. However, in the medical domain, research on region-level LMMs is still limited. BiRD [12] aims to equip MLMMs with grounding and referring capabilities through multi-task learning while maintaining their conversational ability. MAIRA-2 [4] focuses on enhancing LMMs for grounded report generation tasks. Both methods locate specific regions using textual coordinates and rely on single-step diagnoses, lacking the comprehensive perception and reasoning to fully leverage these regions.

**CoT in Medical LMMs** Chain-of-Thought (CoT) prompting guides LLMs through intermediate reasoning steps to solve complex tasks [39, 16]. Recent efforts have introduced CoT into LMMs to enhance visual reasoning. For example, SoM [41] integrates supplementary visual cues (e.g., segmentation maps), while VoCoT [20] and Visual-CoT [33] build instruction datasets for object-centric reasoning. However, CoT integration in medical LMMs remains limited. MedCoT [24] uses Gemini-Pro [35] for CoT generation, but lacks medical domain rigor. MedVLM-R1 [30] applies reinforcement learning for reasoning without reference answers, though it is restricted to multiple-choice VQA. X-Reasoner [25] further shows that without early-stage CoT supervision, RL alone yields suboptimal reasoning, underscoring the need for domain-specific CoT guidance in MLMMs.

## 3 Method

### 3.1 Model Overview

As illustrated in Fig. 2, AOR mainly consists of three components: (i) an image encoder $\mathcal{I}$, responsible for extracting image features; (ii) a region encoder $\mathcal{R}$, deployed to extract multi-scale region features from image features; and (iii) a large language model $\mathcal{LLM}$ is designed to jointly model image, region, and text for reasoning after projecting image and region features into the linguistic space.

### 3.2 Model Development

Fig. 2 (b) shows our three-stage training procedure for AOR. We progressively enable AOR to perform anatomy-centric recognition, detection, reasoning, and report generation. All three training stages use cross-entropy loss for auto-regressive language modeling.

**Stage 1: Anatomical Region Recognition** The first stage aims to align region features with linguistic embeddings, enabling the model to recognize each anatomical region in CXR. During this stage, only the region encoder $\mathcal{R}$ and the region projection $f'_p$ are kept trainable. For each image $I$, we use the anatomical bounding boxes $B = \{(c^j, n^j)_{j=1}^{N_b}\}$ provided by Chest ImaGenome Dataset [40]. Here, $c^j \in \mathbb{R}^4$, $n^j$ and $N_b$ represent the coordinate, region name, and the number of anatomical regions in $I$. The image $I$ is first encoded into the feature maps $\mathbf{z} = \{z_i\}_{i=1}^{N_l}$, where $N_l$ is the number of feature maps. Inspired by GPT4RoI [42], we design the region encoder $\mathcal{R}$ which constructs a hierarchical feature pyramid from four selected layers of the image encoder. According to $c^j$,

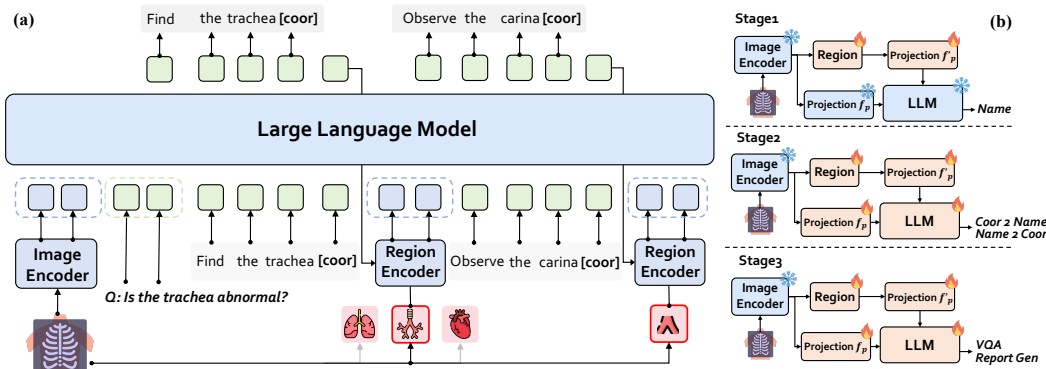

Figure 2: (a) Overview of AOR framework, which flexibly accommodates both textual and optional visual prompts (*i.e.*, region-of-interest (ROI) crops) as input, centered on region-level information to enable multimodal multi-step reasoning; and (b) Three-stage training procedure for AOR.

RoIAlign is applied to generate a $14 \times 14$ feature map from the fused hierarchical feature, followed by a pooling layer to embed multi-scale region features $r^j$. Image projection $f_p$ and region projection $f_p'$ are used to connect $z_{N_l}$ and $r^j$ into the linguistic space. Finally, the $\mathcal{LLM}$ integrates the projected visual features and the text instruction embedding $t$ to recognize the current anatomical region and output its name $n^j$:

$$n^j = \mathcal{LLM}(f_p(z_{N_l}), t, f_p'(r^j)) \tag{1}$$

**Stage 2: Anatomical Region Grounding**  In the second stage, the model is trained to localize anatomical regions, laying the foundation for subsequent reasoning tasks. Since the generation of coordinates requires an overview of the entire image and the generative capability of the LLM, we keep both the LLM and image projection modules trainable. Two types of tasks are considered: (1) To prevent catastrophic forgetting and align the format closer to the reasoning tasks, the model revisits the anatomical region recognition of Stage 1 with some adjustments, i.e., concatenating the coordinates to the region feature. Here, we use bbox $[x_{min}, y_{min}, x_{max}, y_{max}]$ as object coordinates, where $x$ and $y$ are normalized between 0 and 1 relative to the image size. The LLM reads the projected visual features, text instruction embedding $t$, and textual coordinates embedding $c^j$ to predict the region name $n^j$:

$$n^j = \mathcal{LLM}(f_p(z_{N_l}), t, [c^j, f_p'(r^j)]) \tag{2}$$

(2) Given region's name $n^j$, the model grounds the corresponding coordinates $c^j$:

$$c^j = \mathcal{LLM}(f_p(z_{N_l}), t, n^j) \tag{3}$$

**Stage 3: Instruction Tuning**  Based on the pre-trained model, this stage fine-tunes the model using AOR-Instruction (detailed in Section 4) on three tasks:

(1) Medical Visual Question Answering: AOR is capable of handling questions that require both global and local clues, and is flexible enough to accept both textual and optional visual prompts as input. Based on the given prompt, the model centers on anatomical regions to generate logically reasoned answers. During reasoning, each region is represented in a triplet format: ⟨region name⟩ ⟨coordinates⟩ ⟨ROI visual representation⟩, e.g., "svc [0.27, 0.08, 0.92, 0.81] $r_{svc}$". Once the end of the coordinates token "]" is generated, the region encoder $\mathcal{R}$ is activated to obtain the ⟨ROI visual representation⟩ based on the coordinates between "[" and "]", which is formulated as follows:

$$ans^j = \mathcal{LLM}(f_p(z_{N_l}), t, [n^j, c^j, f_p'(r^j)]) \tag{4}$$

(2) Full Image Report Generation: Given a CXR, AOR generates a comprehensive report describing the entire image:

$$report = \mathcal{LLM}(f_p(z_{N_l}), t) \tag{5}$$

(3) Region Report Generation: For a CXR, users provide textual and optional visual prompts specifying the anatomical regions of interest. AOR generates a report sentence $s^j$ specifically related to the designated region $r^j$.

$$s^j = \mathcal{LLM}(f_p(z_{N_l}), t, f_p'(r^j)) \tag{6}$$

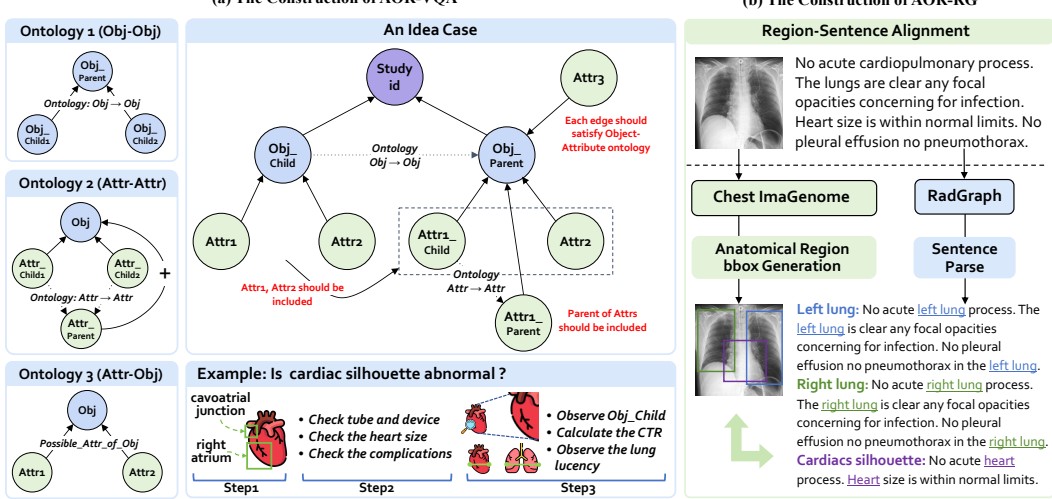

Figure 3: Overview of AOR-Instruction, which consists of two sub-datasets: (a) The construction of AOR-VQA: Anatomical ontologies design → CoT construction → Sample expansion and (b) The construction of AOR-RG: Strict alignment between anatomical region and report sentence.

## 4 Instruction Data

Currently, there is a shortage of high-quality multimodal reasoning datasets for training Medical LMMs, leading to models that lack fine-grained understanding and reasoning capabilities. To bridge this gap, we construct the AOR-Instruction dataset by leveraging anatomical regions and their ontologies. This dataset, enriched with explainable region-level visual information, helps LMMs achieve a clearer understanding of image content. The AOR-Instruction dataset consists of two components: AOR-VQA for Visual Question Answering (VQA) and AOR-RG for full image and region report generation, containing 290k and 532k data pairs, respectively.

### 4.1 AOR-VQA

AOR-VQA is primarily designed to enhance the model's capabilities in medical VQA. We use MIMIC-CXR-VQA [3], a comprehensive dataset with (Image, Question, Answer) samples, as our primary data source. It includes both global-level questions that evaluate the overall findings of a CXR (e.g., "Do you notice any abnormalities?"), and local-level questions that focus on specific anatomical regions (e.g., "Can hyperaeration be detected in the left lung?").

The entire construction process is conducted under the guidance of expert physicians—two board-certified radiologists and one clinician with 24, 18, and 27 years of experience, respectively. We meticulously design and refine three types of anatomical ontologies. Based on these ontologies, we construct a Chain of Thought (CoT) for each sample. Finally, the expanded information is attached to each sample, resulting in a structure that includes (Image, Question, Region Box, CoT Answer). The details are as follows:

#### 4.1.1 Anatomical Ontologies Design

We define anatomical regions in the CXR as objects, each associated with several attributes selected from a pool of 68 attributes across five categories. Fig. 3 (a) illustrates our anatomical ontologies.

**Ontology 1: Hierarchical Relationships Between Objects** In visual perception, humans organize content into hierarchical structures to understand the part-whole relationships within images, thereby obtaining the answers they seek. Fortunately, such part-whole hierarchical relationships clearly exist between anatomical regions. As illustrated in Fig. 3 (a), Obj_Parent (e.g., mediastinum) includes two related objects: Obj_child1 (e.g., upper mediastinum) and Obj_child2 (e.g., cardiac silhouette). By leveraging this hierarchical relationship, the model can engage in reasoning by shifting focus from the whole to the parts and then comprehensively considering the whole based on the parts. Therefore, we explore and organize the hierarchical relationships of 38 anatomical regions.

**Ontology 2: Causal Relationships Between Attributes**   During the progression of specific attributes, multiple conditions can dynamically interrelate and influence one another. Additionally, attributes categorized as anatomical findings are typically the imaging manifestations of attributes classified as diseases. Thus, we can construct causal relationships between different attributes. As shown in Fig. 3 (a), if Attr_Child (e.g., lobar/segmental collapse) exists, then Attr_Parent (e.g., atelectasis) within an Obj must also be present. Leveraging these causal relationships enables the model to better understand the associations between attributes and utilize other attributes to complete the reasoning process when encountering attributes with unclear or difficult-to-determine visual manifestations. Consequently, the causal relationships of 68 attributes are constructed.

**Ontology 3: Restrictive Relationships Between Objects and Attributes**   Finally, we consider the restrictive relationships between objects and attributes. As shown in Fig. 3 (a), only certain attributes appear within specific objects. For example, fractures can never occur at the cardiac silhouette, while pleural effusion most commonly affects the costophrenic angle. By utilizing such restrictive relationships, the model can eliminate certain scenarios, enabling more rational and effortless reasoning. Therefore, the restrictive relationships between 38 objects and 68 attributes are organized. Refer to Appendix A.2.1 for more details about the three Ontologies.

### 4.1.2   CoT construction

Integrating the aforementioned three ontologies, as illustrated in Fig. 3 (a), we organize an ideal case for each image (represented as a Study id node) in the source dataset, establishing connections between all objects and attributes for each sample. For global-based questions, we do not expand the answers, enabling the model to make fine-grained observations while also developing global summarization skills. For local-based questions, we construct a rigorous and comprehensive CoT for each question based on the ideal case, following the steps below.

**Step 1: Identify Sub-objects (Using Ontology 1)**   For the queried object, we first identify its sub-objects (if it is already the smallest anatomical structure unit, this step is skipped). Considering the question: "Is the cardiac silhouette abnormal?", we identify the right atrium and the cavoatrial junction as the sub-objects of the cardiac silhouette for focused analysis.

**Step 2: Consider All Possible Attributes (Using Ontology 2)**   If the question targets a specific attribute, analysis and reasoning are conducted solely for that attribute. However, if the question concerns all abnormalities of the queried object, all possible scenarios must be taken into account. Continuing with the example from Step 1, for abnormalities in the cardiac silhouette contour, this primarily includes the presence of tubes or devices, changes in the size of the cardiac silhouette, and the development of other associated complications.

**Step 3: Associate the Relevant Objects and Attributes (Using Ontology 3)**   Once we have identified the attributes to be discussed, we focus our observation and reasoning on their primary associated objects or sub-objects. Continuing with the example above: for the presence of tubes or devices, particularly observe the right atrium and the cavoatrial junction; for measurements, i.e., cardiac silhouette size, place cardiac silhouette within the global context and compare it to the size of the entire thorax; for the development of complications, after detecting an enlarged cardiac silhouette, consider other features, i.e., pulmonary translucency, to further assess the presence of lung opacity.

Following the three steps outlined above, we can construct a CoT answer for any combination of objects and attributes (or categories and abnormalities). A total of $(68+5+1)\times38 = 2,812$ types of CoT answers are constructed, all of which are reviewed and refined by three expert physicians.

Furthermore, source data includes complex combinatorial questions, such as those involving conjunction and disjunction, which are commonly seen in real-world application scenarios. Therefore, we decouple such data by extracting the involved sub-questions to perform the aforementioned CoT and finally conduct an additional logical inference based on the answers to the sub-questions.

### 4.1.3   Sample Expansion

All samples in the dataset are expanded from (Image, Question, Answer) to (Image, Question, Region Box, CoT Answer) according to the above rules. For more details, please refer to Appendix A.2.1.

Table 1: Comparison of methods on MIMIC-CXR-VQA, VQA-RAD, and CheXpert. "–" means Med-Flamingo and MedVLM-R1 lack training code and thus cannot be fine-tuned on MIMIC-CXR-VQA. "*" means CheXagent lacks training code, but its instruction data include MIMIC-CXR-VQA, so we report zero-shot results. Results in gray are excluded from the comparison since CheXagent data include VQA-RAD. **Bold** numbers mark the best result in each column.

| Method | Res. | MIMIC-CXR-VQA | | | VQA-RAD | | CheXpert | |
|---|---|---|---|---|---|---|---|---|
| | | verify | choose | query | closed | open | closed | open |
| *General-domain LMM* | | | | | | | | |
| LLaVA [23] | $224^2$ | 75.97 | 56.07 | 58.87 | 43.84 | 21.09 | 27.72 | 34.50 |
| LLaVA-1.5 [22] | $336^2$ | 75.25 | 58.70 | 56.10 | 44.74 | 13.54 | 27.72 | 33.88 |
| GPT4RoI [42] | $224^2$ | 77.16 | 56.37 | 60.54 | 43.84 | 17.64 | 52.19 | 32.65 |
| VoCoT [20] | $448^2$ | 76.17 | 48.79 | 60.70 | 35.62 | 21.38 | 49.60 | 40.89 |
| *Medical-domain LMM* | | | | | | | | |
| LLaVA-Med [18] | $224^2$ | 75.71 | 58.31 | 60.37 | 61.64 | 22.36 | 54.95 | 42.08 |
| Med-Flamingo [27] | $224^2$ | - | - | - | 28.77 | 23.89 | 40.99 | 39.80 |
| XrayGPT [36] | $224^2$ | 60.00 | 40.97 | 24.07 | 43.84 | 22.51 | 60.59 | 30.45 |
| CheXagent [9] | $448^2$ | 75.02* | 33.49* | 48.49* | 68.49 | 24.94 | 62.28 | 32.14 |
| MedVLM-R1 [30] | *Dyn* | - | - | - | 52.05 | 22.07 | 60.89 | 33.55 |
| AOR(Ours)-t | $336^2$ | **80.48** | **71.96** | **65.05** | **63.01** | **28.19** | 71.58 | **53.85** |
| AOR(Ours)-r/t | $336^2$ | 80.68 | 70.16 | 65.43 | 57.53 | 24.99 | 74.06 | 45.35 |

## 4.2 AOR-RG

**Full Image Report Generation**    We directly utilize the image-report pairs provided in MIMIC-CXR [15], make use of frontal images, and include findings and impressions in the report.

**Region Report Generation**    To perform fine-grained region report generation, we need to decompose raw report data into fine-grained descriptions for each organ mentioned in medical scans. As shown in Fig. 3 (b), we utilize the bounding boxes provided by Chest ImaGenome Dataset and parse the text using RadGraph [14], employing the rules proposed in ASG [19] to achieve strict alignment between the two. Additionally, we optimize the alignment method to handle cases where two different anatomical regions appear in the same short sentence, introducing new rules to split such sentences into two separate ones. This approach constructs region-sentence pairs for each image-report pair.

# 5 Experiments

## 5.1 Experiment Settings

**Implementation Details**    We initialize the image encoder with CLIP-ViT-L/14 [31] and the language model with LLaVA-1.5 [22]. The image resolution is set to 336×336. We use AdamW as our optimizer, with a learning rate of $2 \times 10^{-5}$. Experiments are conducted using 4 NVIDIA A100 GPUs.

**Dataset**    1) For the VQA task, we evaluate on the test sets of MIMIC-CXR-VQA [3], VQA-RAD [17], and CheXpert [13]. MIMIC-CXR-VQA contains 500 images and 13,793 QA pairs in the test dataset. CheXpert contains 191,229 frontal chest radiographs, and we use the expert-labeled validation set as the test data, which contains 202 images and 1,212 QA pairs. VQA-RAD includes 315 images and 3,515 QA pairs across the head, chest, and abdomen. We filter it to include only chest X-ray images and their corresponding question-answer pairs, resulting in 69 images and 102 QA pairs in the test dataset. 2) For the full image report generation task, we use MIMIC-CXR, a large publicly available dataset of chest radiographs with free-text radiology reports, to evaluate the model's performance. The test dataset contains 500 images and their corresponding reports. 3) For the region report generation task, we use the same 500 images and sample 3 anatomical regions per image for evaluation.

**Evaluation Metrics**    For the VQA task, regarding MIMIC-CXR-VQA, its questions can be categorized into three primary semantic types: For the "verify" questions, which include yes/no questions, we report the accuracy; for the "choose" questions, which involve selection from provided options, we also report the accuracy; for the "query" questions, where the answers are in the form of a list, we report the $F_1$ score (micro). For CheXpert and VQA-RAD, following [18], for closed-end

Table 2: Comparison of report generation methods on MIMIC-CXR. "†" means Res. is $336^2$.

| Method | MIMIC-CXR | | |
|---|---|---|---|
| | R-L | BERTScore | $F_1$CheXbert |
| *Full image report generation* | | | |
| LLaVA-Med | 13.49 | 75.93 | 0.40 |
| Med-Flamingo | 5.21 | 71.26 | 13.61 |
| XrayGPT | 24.02 | 83.18 | 26.71 |
| CheXagent | 24.09 | 83.52 | 37.54 |
| LLaVA-Rad† | 24.22 | 83.67 | 46.20 |
| AOR(Ours)-t | **25.37** | **83.92** | **46.53** |
| AOR(Ours)-r/t | 25.38 | 83.95 | 48.28 |
| *Region report generation* | | | |
| LLaVA-Med | 9.47 | 71.51 | 16.70 |
| Med-Flamingo | 12.98 | 75.24 | 14.66 |
| XrayGPT | 15.80 | 80.19 | 19.96 |
| CheXagent | 20.79 | 81.57 | 33.02 |
| LLaVA-Rad† | 18.71 | 81.80 | 31.40 |
| AOR(Ours)-t | **35.11** | **84.54** | **36.65** |
| AOR(Ours)-r/t | 35.62 | 84.76 | 36.89 |

Table 3: Comparison of the different CoT representations.

| ID | coor | region | CoT | verify | choose | query |
|---|---|---|---|---|---|---|
| 1 | ✗ | ✗ | ✗ | 76.83 | 61.07 | 58.77 |
| 2 | ✗ | ✗ | ✓ | **80.69** | 67.62 | 63.37 |
| 3 | ✓ | ✗ | ✓ | 79.14 | 69.54 | 63.89 |
| 4 | ✓ | ✓ | ✓ | 80.68 | **70.16** | **65.43** |

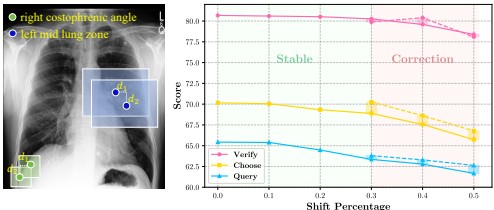

Figure 4: Impact of anatomical region shifts on model predictions. Solid lines represent the original visual prompt bboxes, dashed lines represent bboxes enlarged by 10% of the original size.

questions with a single correct answer, we report accuracy; for open-end questions, we use recall to evaluate the model's responses. For the report generation task, we selected ROUGE-L (R-L) [21], BERTScore [43], and $F_1$CheXbert [26] as evaluation metrics to compare model performance at the word level, semantic level, and clinical efficacy level.

## 5.2 Quantitative Comparison

**Performance on MIMIC-CXR-VQA**    We fine-tune all models on MIMIC-CXR-VQA. For a fair comparison, we introduce AOR-t, which uses the same training data as baseline methods, where all questions are in textual form. AOR-r/t represents a setting where questions are presented in both textual and visual formats during training and inference, enabling multimodal interaction. This reflects a more general and practical usage scenario, requiring the model to dynamically parse and reason over heterogeneous input modalities. Notably, regardless of how the questions are presented, both AOR-t and AOR-r/t leverage region-level visual information to enhance reasoning throughout the answering process. As revealed in Table 1, AOR-t outperforms the second-best method by an average of 6.98%, especially on the more complex "choose" and "query" types, highlighting the advantage of reasoning centered on anatomical regions.

**Zero-shot Transfer to VQA-RAD and CheXpert**    We evaluate AOR's generalization ability on unseen data distributions, i.e., VQA-RAD and CheXpert. As shown in Table 1, AOR-t obtains superior performance on both datasets. Moreover, AOR is capable of reasoning and providing logical answers rather than simply responding with "yes" or "no".

**Performance on MIMIC-CXR**    Table 2 shows that AOR outperforms all previous methods in both full-image and region-based report generation. Notably, it addresses the gap in generating report sentences for specific regions, which remains a limitation in previous MLMMs.

## 5.3 Ablation Studies and Discussions

**Comparison of CoT Representations**    As shown in Table 3, we demonstrate the effectiveness of the AOR representation by comparing it with three alternative CoT representations for the VQA task. ID-1 uses the original answer without any CoT reasoning, resulting in the lowest accuracy, especially on complex question types. ID-2 introduces a text-based CoT that outlines object-level reasoning steps, which significantly improves performance and highlights the value of explicit multi-step reasoning. ID-3 adds textual coordinates, enabling the model to gain spatial awareness by first identifying key anatomical regions. Building on this, ID-4, our proposed AOR representation, further integrates cropped, multi-scale visual features from each ROI. This enhancement allows the model to leverage direct visual cues during reasoning and achieves the best overall performance.

Table 4: Comparison of AOR's referring and grounding capabilities under different training strategies.

| Strategy | | VQA | | | Referring | Grounding | | |
| Stage 1 | Stage 2 | verify | choose | query | Acc | R@0.3 | R@0.5 | R@0.7 |
| --- | --- | --- | --- | --- | --- | --- | --- | --- |
| ✗ | ✗ | 79.53 | 67.96 | 62.33 | 98.03 | 98.48 | 96.16 | 88.79 |
| ✓ | ✗ | 80.53 | 70.05 | 63.96 | 98.30 | 98.31 | 96.54 | 89.40 |
| ✓ | ✓ | **80.68** | **70.16** | **65.43** | **98.58** | **98.56** | **96.62** | **90.40** |

---

**Medical Visual Question Answering**

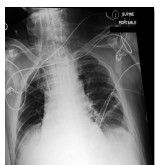

***Question:*** Can both enteric tube and IJ line be identified within the mediastinum?    ***Answer:*** Yes

***AOR:*** Find the mediastinum [0.32, 0.12, 0.63, 0.60]. For the enteric tube, observe the upper mediastinum [0.36, 0.18, 0.57,0.36], the cardiac silhouette [0.32, 0.37, 0.63, 0.60], and the mediastinum [0.32, 0.12, 0.63, 0.60]. Considering all observations, there is enteric tube in the mediastinum [0.32, 0.12, 0.63, 0.60]. For the IJ line, observe the svc [0.36, 0.25, 0.48, 0.36], the cavoatrial junction [0.32, 0.37, 0.44, 0.43], the right atrium [0.32, 0.43, 0.44, 0.56], and the mediastinum [0.32,0.12,0.63,0.60]. Considering all observations, there is IJ line in the mediastinum [0.32, 0.12, 0.63, 0.60]. So the answer is yes.

**Full Image Report Generation & Region Report Generation**

***GroundTruth:*** AP portable upright view of the chest. ET tube appears in stable position with its tip 4 cm above the carina. NG tube courses into the left upper abdomen. Mild basal atelectasis noted. Otherwise lungs are clear. Cardiomediastinal silhouette is stable. No acute bony abnormalities. Tubes positioned appropriately. Mild bibasilar atelectasis.

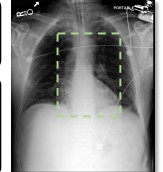

***AOR:*** Endotracheal tube tip terminates approximately 5 cm from the carina. An enteric tube tip is within the stomach. Heart size is normal. Mediastinal and hilar contours are unremarkable. Pulmonary vasculature is not engorged. Patchy opacities are noted in the lung bases, potentially atelectasis. No pleural effusion or pneumothorax is present. No acute osseous abnormality is detected. Standard positioning of the endotracheal and enteric tubes.

***Question:*** Please generate a corresponding radiology report based on this region of the Chest X-ray.
***GroundTruth:*** cardiomediastinal silhouette is stable .

***AOR:*** the cardiomediastinal silhouette is within normal limits.

Figure 5: Qualitative analysis of AOR on VQA task and report generation task.

**Referring and Grounding Capabilities**    We analyze the model's referring and grounding capabilities to verify the rationality of the CoT process. In Table 4, when all three training stages are involved, the model achieves a 98.58% referring accuracy and 90.40% R@0.7, ensuring both the accuracy and explainability of the answers. When Stage 2 is removed, the recall drops accordingly, indicating that the grounding task in Stage 2 lays a solid foundation for reasoning. Similarly, without Stage 1, the performance of referring is affected, demonstrating the effectiveness of our three-stage training.

**The Impact of Anatomical Region Shifts on Model Predictions**    In practical applications, considering that radiologists may not provide perfectly accurate region prompts and patients might offer bboxes that deviate from standard anatomical regions, we explore the impact of bbox shifts on prediction accuracy. As shown in Fig. 4, we apply shifts to the bbox in the horizontal and vertical directions during model inference ($d_1 = r \cdot w, d_2 = r \cdot h$), where $r$ is a random number between 0 and percentage $p \in 0.1 \cdot \{0, \ldots, 5\}$, and $w$ and $h$ are the width and height of the image. When $p \in [0.1, 0.2]$, the model's performance is barely affected. However, a downward trend emerges when $p$ exceeds 0.3. This indicates that for shifts that do not impact the bbox class prediction, AOR is robust enough to produce correct predictions. Conversely, when the shift becomes larger, the model might misclassify the category during the first step of reasoning (e.g., a shifted left lung might be misclassified as the mediastinum), thereby affecting subsequent accuracy. This also indirectly demonstrates that our model performs step-by-step logical analysis of anatomical regions during reasoning. As shown by the dashed lines in the Fig. 4, we explore the impact of moderately enlarging the bboxes in the visual prompt (by 10% of the original size). This brought a certain degree of performance gain, but there remains a performance gap compared to using perfectly accurate bboxes.

**Scalability of CoT Construction**    We generate CoT answers for each sample in VQA-RAD to verify the generalizability of our construction method to other CXR datasets. The construction is accomplished using a two-stage approach, Keyword Mapping and Sample Expansion. Please refer to Appendix B.5 for more details. As shown in Table 5, the model fine-tuned with CoT answers achieves a performance gain of 4.78%.

Table 5: Performance gain from CoT answers on VQA-RAD.

| | 1 Epoch | | 3 Epochs | |
| | Closed | Open | Closed | Open |
| --- | --- | --- | --- | --- |
| w/o CoT | 66.38 | 32.56 | 83.62 | 59.33 |
| w/ CoT | 68.97 | 37.16 | 89.66 | 65.23 |

## 5.4 Qualitative Analysis

Fig. 5 illustrates AOR's capabilities in medical grounded chat and referential dialogue. For the VQA task, AOR is capable of generating correct and logically reasoned answers. For the report generation task, due to the incorporation of fine-grained anatomical regions, AOR demonstrates a stronger grasp of details, such as ET tube, NG tube, and basal atelectasis. Moreover, it can generate corresponding report sentences for specified regions. Please refer to Appendix C for more cases.

## 6 Conclusion

In this paper, we empower MLMMs with anatomy-centric reasoning capabilities, by (1) proposing the AOR framework, which centers on the anatomical regions relevant to the given question, and integrating the regions' positional and representational information to conduct multimodal multi-step reasoning; (2) developing the medical CoT dataset AOR-Instruction, which provides tailored CoT answers for each VQA sample and strictly aligned region-sentence pairs for report generation. Experiments demonstrate the superiority of AOR over prior MLMMs in visual question answering, report generation, referring, and grounding, revealing its potential in clinical practice.

## Acknowledgements

This work was supported by Shanghai Natural Science Foundation (No. 25ZR1401028), and the Science and Technology Commission of Shanghai Municipality (No. 23511100602).

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

# Appendix

# A Data Details

## A.1 Data Source

Table A1: Summary of datasets and tasks across different stages.

| Stages | Tasks | Source | Size |
|---|---|---|---|
| Stage 1 | Anatomical Region Recognition | Chest ImaGenome | 133k × 3 |
| Stage 2 | Anatomical Region Recognition | Chest ImaGenome | 133k × 3 |
| | Anatomical Region Grounding | Chest ImaGenome | 133k × 3 |
| Stage 3 | Visual Question Answer | MIMIC-CXR-VQA | 290k |
| | Full Image Report Generation | MIMIC-CXR | 133k |
| | Region Report Generation | MIMIC-CXR | 133k × 3 |

We present our data details in Table A1. We utilize the dataset under the PhysioNet license, and the following is a detailed introduction to it:

- **MIMIC-CXR**[1]    A large publicly available dataset of chest radiographs with free-text radiology reports. The dataset contains 377,110 images corresponding to 227,835 radiographic studies performed at the Beth Israel Deaconess Medical Center in Boston, MA.

- **Chest ImaGenome Dataset**[2]    Chest ImaGenome dataset constructs fine-grained annotations on top of MIMIC-CXR. The annotations for each CXR are structured as an anatomy-centered scene graph. Based on this scene graph, pairs of anatomical regions and report sentences can be derived. The dataset size is consistent with that of MIMIC-CXR.

- **MIMIC-CXR-VQA**[3]    MIMIC-CXR-VQA is created based on the fine-grained annotations of the Chest ImaGenome dataset, establishing a medical visual question answering (VQA) benchmark. It includes a variety of questions, such as those about the entire image and those focused on one or multiple anatomical regions. The questions are diverse and challenging.

Since the three data sources are interrelated (MIMIC-CXR → Chest ImaGenome Dataset→ MIMIC-CXR-VQA), MIMIC-CXR-VQA's data selection and split strategy are adopted across all three to prevent data leakage, with 133k images for training, 500 images for testing. In Stage 1 and Stage 2, we use the anatomical bounding boxes provided by the Chest ImaGenome dataset, randomly selecting three anatomical regions per image for training. Similarly, in Stage 3, during region report generation, three anatomical regions are randomly selected per image for generating corresponding report sentences. Additionally, as shown in Fig. A1, we analyze the distribution of different types of questions in MIMIC-CXR-VQA.

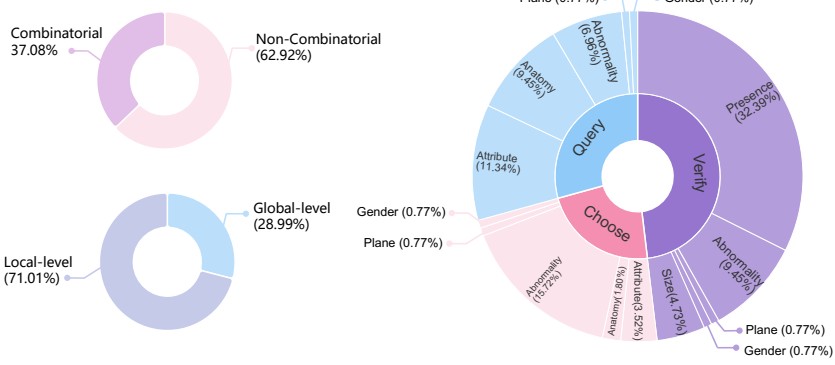

Figure A1: Distribution of question types in MIMIC-CXR-VQA.

---

[1]MIMIC-CXR: Available at https://physionet.org/content/mimic-cxr-jpg/2.0.0/

[2]Chest ImaGenome Dataset: Available at https://physionet.org/content/chest-imagenome/1.0.0/

[3]MIMIC-CXR-VQA: Available at https://physionet.org/content/mimic-ext-mimic-cxr-vqa/1.0.0/

## A.2 Instruction Data Construction

The AOR-Instruction consists of two parts: AOR-VQA and AOR-RG.

### A.2.1 AOR-VQA

Table A2 displays the objects, attributes, and categories used in AOR-VQA, including 36 objects and 68 attributes (across 5 categories). We adopt the same values as in MIMIC-CXR-VQA and adjust the capitalization of the specialized terms IJ line and PICC.

Table A2: The objects, attributes, and categories used in AOR-VQA.

|  | Values |
|---|---|
| object | abdomen, aortic arch, cardiac silhouette, carina, cavoatrial junction, left apical zone, left breast, left chest wall, left clavicle, left costophrenic angle, left hemidiaphragm, left hilar structures, left lower lung zone, left lung, left mid lung zone, left shoulder, left upper lung zone, mediastinum, neck, right apical zone, right atrium, right breast, right chest wall, right clavicle, right costophrenic angle, right hemidiaphragm, right hilar structures, right lower lung zone, right lung, right mid lung zone, right shoulder, right upper lung zone, spine, svc, trachea, upper mediastinum |
| attribute | *(anatomical finding)* lung opacity, airspace opacity, consolidation, infiltration, atelectasis, linear/patchy atelectasis, lobar/segmental collapse, pulmonary edema/-hazy opacity, vascular congestion, vascular redistribution, increased reticular markings/ild pattern, pleural effusion, costophrenic angle blunting, pleural/parenchymal scarring, enlarged cardiac silhouette, mediastinal displacement, mediastinal widening, enlarged hilum, tortuous aorta, vascular calcification, pneumomediastinum, pneumothorax, hydropneumothorax, lung lesion, mass/nodule (not otherwise specified), multiple masses/nodules, calcified nodule, superior mediastinal mass/enlargement, rib fracture, clavicle fracture, spinal fracture, hyperaeration, cyst/bullae, elevated hemidiaphragm, sub-diaphragmatic air, subcutaneous air, hernia, scoliosis, spinal degenerative changes, shoulder osteoarthritis, bone lesion, *(disease)* pneumonia, fluid overload/heart failure, copd/emphysema, granulomatous disease, interstitial lung disease, goiter, lung cancer, aspiration, alveolar hemorrhage, pericardial effusion, *(device)* cabg grafts, prosthetic valve, cardiac pacer and wires, *(technical assessment)* low lung volumes, rotated, breast/nipple shadows, *(tubes and lines)* chest tube, mediastinal drain, endotracheal tube, tracheostomy tube, PICC, IJ line, chest port, subclavian line, swan-ganz catheter, intra-aortic balloon pump, enteric tube |
| category | anatomical finding, device, disease, technical assessment, tubes and lines |

Fig. A2 illustrates the step-by-step construction process of AOR-VQA, which consists of ontologies design, chain-of-thought (CoT) construction, and sample expansion to enrich dataset annotations.

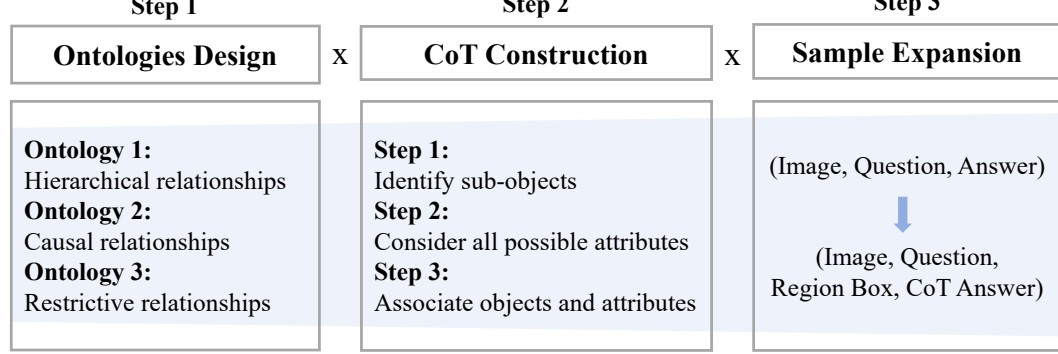

Figure A2: Overview of the AOR-VQA construction process.

**Step 1: Ontologies Design**    Fig. A3 illustrates the hierarchical relationships between objects (Ontology 1) that we utilize, where the green boxes represent objects existing in the silver/gold datasets of the Chest Imagenome Dataset, and the gray boxes represent the objects we used during the construction process.

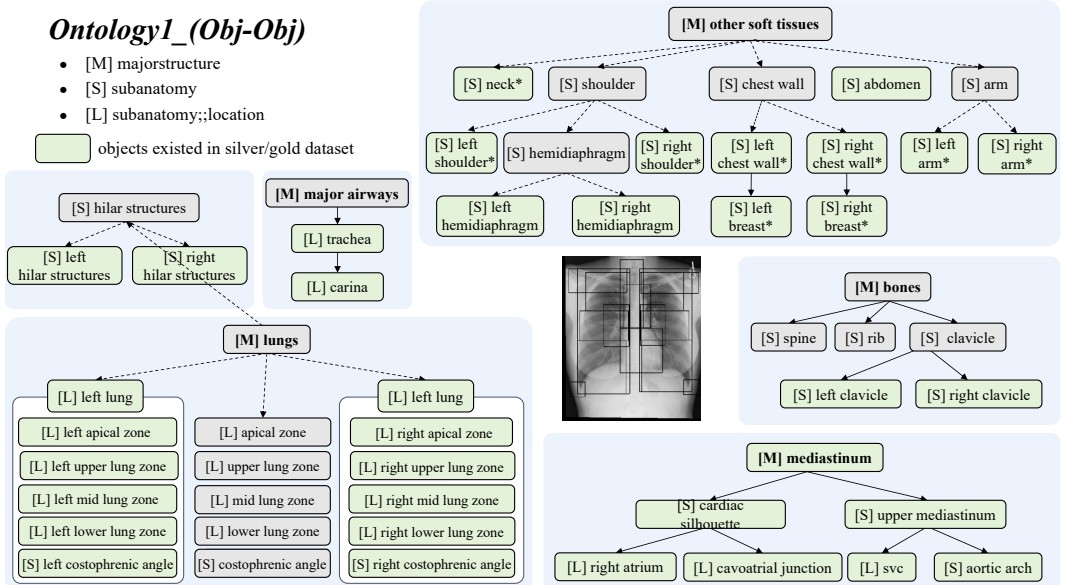

Figure A3: Ontology 1: Hierarchical relationships between objects.

Table A3 and Table A4 display the causal relationships between attributes (Ontology 2) and the restrictive relationships between objects and attributes (Ontology 3).

Table A3: Ontology 2: Causal relationships between attributes (Child → Parent(s))

| Attribute (Child) | Attribute(s) (Parent) |
|---|---|
| airspace opacity | lung opacity |
| alveolar hemorrhage | lung opacity |
| aspiration | lung opacity |
| atelectasis | lung opacity |
| bronchiectasis | lung opacity |
| calcified nodule | lung opacity, lung lesion |
| consolidation | lung opacity |
| costophrenic angle blunting | lung opacity, pleural effusion |
| fluid overload/heart failure | lung opacity |
| granulomatous disease | lung opacity |
| hydropneumothorax | lung opacity, pneumothorax, pleural effusion |
| increased reticular markings/ild pattern | lung opacity |
| infiltration | lung opacity |
| interstitial lung disease | lung opacity |
| linear/patchy atelectasis | atelectasis, lung opacity |
| lobar/segmental collapse | atelectasis, lung opacity |
| lung cancer | lung opacity |
| lung lesion | lung opacity |
| mass/nodule (not otherwise specified) | lung opacity, lung lesion |
| multiple masses/nodules | lung opacity, lung lesion |
| pleural effusion | lung opacity |
| pleural/parenchymal scarring | lung opacity |

*Continued on next page*

| Child Attribute | Parent Attribute(s) |
|---|---|
| pneumonia | lung opacity |
| pulmonary edema/hazy opacity | lung opacity |
| vascular congestion | lung opacity |

Table A4: Ontology 3: Restrictive relationships between objects (Parents) and attributes (Children).

| Object (Parent) | Attributes (Children) |
|---|---|
| cardiac silhouette | cabg grafts, aortic graft/repair, prosthetic valve, cardiac pacer and wires, fluid overload/heart failure, pericardial effusion, lung opacity, mediastinal displacement, pneumomediastinum, enlarged cardiac silhouette, vascular calcification, hernia, low lung volumes, rotated, enteric tube, mediastinal drain, pigtail catheter, chest port, IJ line, picc, subclavian line, swan-ganz catheter |
| mediastinum | cabg grafts, aortic graft/repair, prosthetic valve, cardiac pacer and wires, lung cancer, fluid overload/heart failure, pericardial effusion, goiter, calcified nodule, lung opacity, lung lesion, mediastinal displacement, mediastinal widening, pneumomediastinum, superior mediastinal mass/enlargement, enlarged cardiac silhouette, tortuous aorta, vascular calcification, hernia, low lung volumes, rotated, endotracheal tube, tracheostomy tube, enteric tube, mediastinal drain, pigtail catheter, chest port, IJ line, intra-aortic balloon pump, picc, subclavian line, swan-ganz catheter |
| aortic arch | aortic graft/repair, mediastinal widening, tortuous aorta, vascular calcification, intra-aortic balloon pump |
| upper mediastinum | aortic graft/repair, cardiac pacer and wires, lung cancer, goiter, calcified nodule, lung opacity, lung lesion, mediastinal displacement, mediastinal widening, pneumomediastinum, superior mediastinal mass/enlargement, tortuous aorta, vascular calcification, rotated, endotracheal tube, tracheostomy tube, enteric tube, mediastinal drain, chest port, IJ line, intra-aortic balloon pump, picc, subclavian line, swan-ganz catheter |
| cavoatrial junction | cardiac pacer and wires, chest port, IJ line, picc, subclavian line, swan-ganz catheter |
| arms (left/right) | cardiac pacer and wires, bone lesion, subcutaneous air, skin fold, picc |
| chest wall (left/right) | cardiac pacer and wires, bone lesion, rib fracture, subcutaneous air, breast/nipple shadows, skin fold, chest tube, pigtail catheter, chest port |
| clavicle (left/right) | cardiac pacer and wires, bone lesion, clavicle fracture, shoulder osteoarthritis, rotated, endotracheal tube, tracheostomy tube, chest port, picc, subclavian line |
| right atrium | cardiac pacer and wires, chest port, IJ line, picc, subclavian line, swan-ganz catheter |
| svc | cardiac pacer and wires, chest port, IJ line, picc, subclavian line, swan-ganz catheter |
| lower lung zone (left-/right) | alveolar hemorrhage, aspiration, copd/emphysema, granulomatous disease, interstitial lung disease, pneumonia, lung cancer, fluid overload/heart failure, airspace opacity, calcified nodule, consolidation, cyst/bullae, hyperaeration, increased reticular markings/ild pattern, infiltration, atelectasis, linear/patchy atelectasis, lobar/segmental collapse, lung opacity, lung lesion, mass/nodule (not otherwise specified), multiple masses/nodules, pulmonary edema/hazy opacity, hydropneumothorax, pleural/parenchymal scarring, pneumothorax, vascular congestion, vascular redistribution, bronchiectasis, pleural effusion, breast/nipple shadows, low lung volumes, chest tube, pigtail catheter |

*Continued on next page*

| Object (Parent) | Attributes (Children) |
|---|---|
| lung (left/right) | alveolar hemorrhage, aspiration, copd/emphysema, granulomatous disease, interstitial lung disease, pneumonia, lung cancer, fluid overload/heart failure, airspace opacity, calcified nodule, consolidation, cyst/bullae, hyperaeration, increased reticular markings/ild pattern, infiltration, atelectasis, linear/patchy atelectasis, lobar/segmental collapse, lung opacity, lung lesion, mass/nodule (not otherwise specified), multiple masses/nodules, pulmonary edema/hazy opacity, enlarged hilum, hydropneumothorax, pleural/parenchymal scarring, pneumothorax, vascular congestion, vascular redistribution, bronchiectasis, pneumomediastinum, vascular calcification, costophrenic angle blunting, pleural effusion, breast/nipple shadows, low lung volumes, rotated, endotracheal tube, chest tube, pigtail catheter, chest port, subclavian line, swan-ganz catheter |
| mid lung zone (left-/right) | alveolar hemorrhage, aspiration, copd/emphysema, granulomatous disease, interstitial lung disease, pneumonia, lung cancer, fluid overload/heart failure, airspace opacity, calcified nodule, consolidation, cyst/bullae, hyperaeration, increased reticular markings/ild pattern, infiltration, atelectasis, linear/patchy atelectasis, lobar/segmental collapse, lung opacity, lung lesion, mass/nodule (not otherwise specified), multiple masses/nodules, pulmonary edema/hazy opacity, hydropneumothorax, pleural/parenchymal scarring, pneumothorax, vascular congestion, vascular redistribution, bronchiectasis, pleural effusion, low lung volumes, chest tube, pigtail catheter |
| upper lung zone (left-/right) | alveolar hemorrhage, aspiration, copd/emphysema, granulomatous disease, interstitial lung disease, pneumonia, lung cancer, fluid overload/heart failure, airspace opacity, calcified nodule, consolidation, cyst/bullae, hyperaeration, increased reticular markings/ild pattern, infiltration, atelectasis, linear/patchy atelectasis, lobar/segmental collapse, lung opacity, lung lesion, mass/nodule (not otherwise specified), multiple masses/nodules, pulmonary edema/hazy opacity, hydropneumothorax, pleural/parenchymal scarring, pneumothorax, vascular congestion, vascular redistribution, bronchiectasis, pleural effusion, low lung volumes, chest tube, pigtail catheter, chest port, subclavian line |
| apical zone (left-/right) | aspiration, granulomatous disease, pneumonia, lung cancer, calcified nodule, cyst/bullae, lung opacity, lung lesion, mass/nodule (not otherwise specified), multiple masses/nodules, hydropneumothorax, pleural/parenchymal scarring, pneumothorax, pleural effusion, chest tube, pigtail catheter |
| hilar structures (left-/right) | aspiration, copd/emphysema, granulomatous disease, pneumonia, lung cancer, fluid overload/heart failure, airspace opacity, calcified nodule, consolidation, cyst/bullae, infiltration, atelectasis, linear/patchy atelectasis, lung opacity, lung lesion, mass/nodule (not otherwise specified), multiple masses/nodules, pulmonary edema/hazy opacity, enlarged hilum, pleural/parenchymal scarring, vascular congestion, vascular redistribution, bronchiectasis, pneumomediastinum, vascular calcification, endotracheal tube, swan-ganz catheter |
| hemidiaphragm (left-/right) | copd/emphysema, hyperaeration, elevated hemidiaphragm, diaphragmatic eventration (benign), hernia, sub-diaphragmatic air, enteric tube |
| costophrenic angle (left/right) | lung cancer, fluid overload/heart failure, airspace opacity, calcified nodule, consolidation, infiltration, atelectasis, linear/patchy atelectasis, lung opacity, lung lesion, mass/nodule (not otherwise specified), multiple masses/nodules, pulmonary edema/hazy opacity, hydropneumothorax, pleural/parenchymal scarring, pneumothorax, costophrenic angle blunting, pleural effusion, pigtail catheter |
| neck | goiter, subcutaneous air, endotracheal tube, tracheostomy tube, enteric tube, IJ line, swan-ganz catheter |
| trachea | goiter, mediastinal displacement, endotracheal tube, tracheostomy tube |

*Continued on next page*

Table A4 – Continued from previous page

| Object (Parent) | Attributes (Children) |
|---|---|
| shoulder (left/right) | bone lesion, shoulder osteoarthritis, skin fold, picc |
| spine | bone lesion, scoliosis, spinal degenerative changes, spinal fracture, rotated |
| abdomen | sub-diaphragmatic air, enteric tube, intra-aortic balloon pump, swan-ganz catheter |
| breast (left/right) | breast/nipple shadows |
| carina | endotracheal tube, tracheostomy tube |

**Step 2: CoT Construction**    As shown in Fig. A4, provide a tree diagram to illustrate the CoT construction process. ① First, determine whether the current problem contains conjunctions and disjunctions. If it does, split it into two non-combinatorial samples, analyze each separately, and then combine the results. ② For non-combinatorial samples, assess whether they are global-based or local-based. If they are global-based, directly assign the CoT answer. Otherwise, proceed to step 3. ③ For local-based questions, determine whether the involved object is the smallest unit (indicated by [L] in Fig. A3). If it is, directly assign the CoT answer; otherwise, use Ontology 1 to identify the sub-object. ④ For the current problem, if it focuses on a specific attribute, construct the corresponding CoT answer in conjunction with Ontology 3 under the guidance of expert doctors. If it focuses on category and abnormality, it is necessary to subdivide into sub-attributes to complete the construction of the CoT answer.

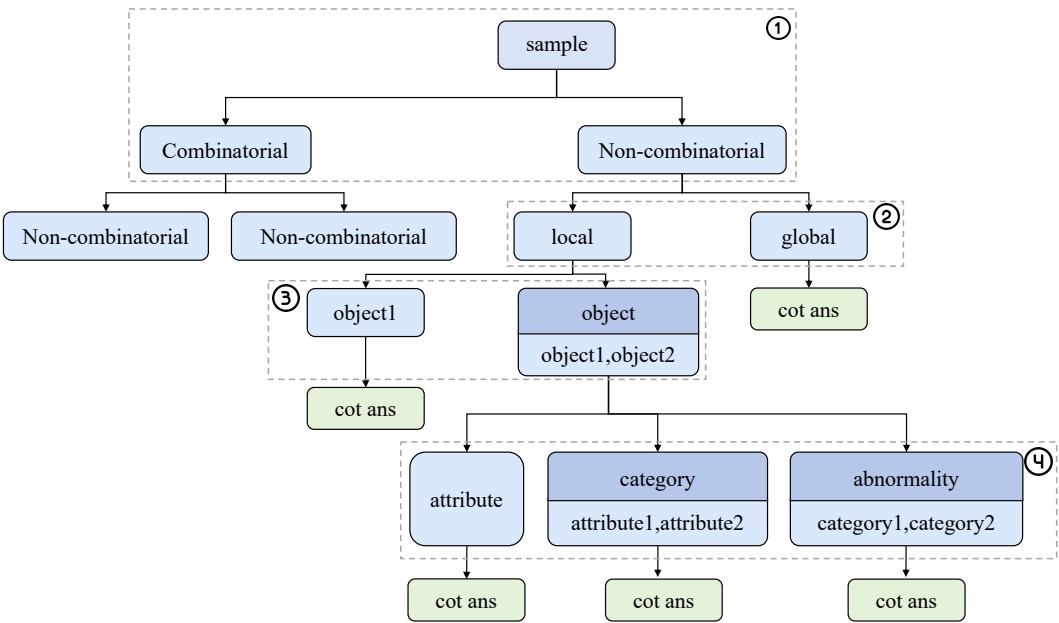

Figure A4: The CoT construction process, illustrating the hierarchical decision-making steps from sample classification to CoT answer generation.

**Step 3: Sample Expansion**  from (Image, Question, Answer) to (Image, Question, Region Box, CoT answer)

The above is the complete construction process of AOR-VQA. Below, we provide examples for different objects. Here, <cot_bbox_*num*>in the examples will be replaced with the corresponding bounding box coordinates in the instruction data.

▶ **cardiac silhouette**  As shown in Table A5, we explain "enlarged cardiac silhouette" and "lung opacity" together: For enlarged cardiac silhouette, reasoning is performed by measuring and calculating the cardiothoracic ratio. For lung opacity, it also involves findings related to the enlarged cardiac silhouette. However, beyond this, further assessment of the cardiac shape and lung lucency is required. For example, the cardiac silhouette maintains its regular shape, suggesting fluid overload/heart failure. Additionally, decreased lung lucency further indicates the presence of lung opacity.

We group "low lung volumes" and "rotated" together as well, as they share a common finding: the cardiac silhouette is not located exactly in the center of the entire thorax. Therefore, further observation of clavicle symmetry is needed to make the final judgment. If the clavicles are symmetrical, it suggests low lung volumes. Conversely, if asymmetry is observed, it indicates rotated, as the entire body appears asymmetrical in this case.

Table A5: Examples of CoT for cardiac silhouette.

| attributes | CoT |
| --- | --- |
| enlarged cardiac silhouette | Find the cardiac silhouette <cot_bbox_1>. Measure the maximal horizontal diameter of the cardiac silhouette <cot_bbox_1>and the maximal horizontal diameter of the thorax <cot_merge_bbox_1>. The cardiothoracic ratio is more than 50%, indicating there is enlarged cardiac silhouette in the cardiac silhouette <cot_bbox_1>. |
| lung opacity | Find the cardiac silhouette <cot_bbox_1>. Measure the maximal horizontal diameter of the cardiac silhouette <cot_bbox_1>and the maximal horizontal diameter of the thorax <cot_merge_bbox_1>. Observe the cardiac silhouette's <cot_bbox_1>shape. Observe the lung lucency <cot_merge_bbox_1>. Considering all observations, there is lung opacity in the cardiac silhouette <cot_bbox_1>. So the answer is yes. |
| low lung volumes | Find the cardiac silhouette <cot_bbox_1>. Observe the position of the cardiac silhouette <cot_bbox_1>within the entire thorax <cot_merge_bbox_1>. The cardiac silhouette <cot_bbox_1>is displaced, and the left clavicle <cot_bbox_16>and right clavicle <cot_bbox_17>are symmetrical. Therefore, there are low lung volumes in the cardiac silhouette <cot_bbox_1>. |
| rotated | Find the cardiac silhouette <cot_bbox_1>. Observe the position of the cardiac silhouette <cot_bbox_1>within the entire thorax <cot_merge_bbox_1>. The cardiac silhouette <cot_bbox_1>is displaced, and the left clavicle <cot_bbox_16>and right clavicle <cot_bbox_17>are asymmetric. Therefore, there is rotated in the cardiac silhouette <cot_bbox_1>. |

▶ **mediastinum**  For the mediastinum, Table A6 presents three examples related to tubes and lines to demonstrate the reasonable and clinically relevant reasoning process of AOR-VQA.

For the IJ line, it typically passes through the SVC and the cavoatrial junction (the recommended tip position for the IJ line). Here, we also emphasize observing the right atrium to ensure the catheter's position is correct and does not extend beyond the cavoatrial junction. Unlike the IJ line, the enteric tube is a medical device inserted through the nose or mouth into the digestive tract. Since the enteric tube traverses almost the entire chest X-ray, we cannot focus solely on the svc and cavoatrial junction. Instead, we observe from the upper mediastinum to the cardiac silhouette. Finally, the endotracheal tube is a flexible plastic tube inserted into the trachea through the mouth or nose to maintain an open airway. Thus, we primarily focus on the upper mediastinum for evaluation.

These reasoning methods for different tubes and lines enable the model to understand each tube/line's position and function effectively. This capability is reflected in our report generation performance (please refer to Fig. 5 in the main manuscript).

Table A6: Examples of CoT for mediastinum.

| attributes | CoT |
|---|---|
| IJ line | Find the mediastinum <cot_bbox_4>. Observe the svc <cot_bbox_7>, the cavoatrial junction <cot_bbox_2>, the right atrium <cot_bbox_3>, and the mediastinum <cot_bbox_4>. Considering all observations, there is IJ line in the mediastinum <cot_bbox_4>. So the answer is yes. |
| enteric tube | Find the mediastinum <cot_bbox_4>. Observe the upper mediastinum <cot_bbox_5>, the cardiac silhouette <cot_bbox_1>, and the mediastinum <cot_bbox_4>. Considering all observations, there is enteric tube in the mediastinum <cot_bbox_4>. |
| endotracheal tube | Find the mediastinum <cot_bbox_4>. Observe the upper mediastinum <cot_bbox_5>, and there is endotracheal tube in the upper mediastinum <cot_bbox_5>. So the answer is yes. |

▶ **left lung** The left lung is a relatively large region, making direct judgment challenging. Thanks to Ontology 3, for specific attributes, we can focus on certain sub-objects. As shown in Table A7, for breast/nipple shadows, we only need to locate the left upper lung zone; for pleural effusion, we specifically observe the left costophrenic angle.

Table A7: Examples of CoT for left lung.

| attributes | CoT |
|---|---|
| breast/nipple shadows | Find the left lung <cot_bbox_8>. Observe the left upper lung zone <cot_bbox_9>, and there are breast/nipple shadows in the left upper lung zone <cot_bbox_9>. So the answer is yes. |
| pleural effusion | Find the left lung <cot_bbox_8>. Observe the left upper lung zone <cot_bbox_9>, the left mid lung zone <cot_bbox_10>, the left lower lung zone <cot_bbox_11>, and the left lung <cot_bbox_8>. Pay special attention to the left costophrenic angle <cot_bbox_20>. Considering all observations, there is pleural effusion in the left lung <cot_bbox_8> |

Table A8 presents examples of compositional questions. Compositional questions are broken down into sub-questions, reasoned step by step, and finally concluded with a judgment.

Table A8: Examples of compositional questions.

*Question*: Are there indications of both vascular calcification and mediastinal displacement in the upper mediastinum?
*CoT answer*: Find the upper mediastinum <cot_bbox_5>. For the vascular calcification, observe the aortic arch <cot_bbox_6>, and there is vascular calcification in the aortic arch <cot_bbox_6>. Therefore, there is vascular calcification in the upper mediastinum <cot_bbox_5>. For the mediastinal displacement, observe the position of the upper mediastinum <cot_bbox_5>within the entire thorax <cot_merge_bbox_1>. The upper mediastinum <cot_bbox_5>is displaced, indicating there is mediastinal displacement in the upper mediastinum <cot_bbox_5>. So the answer is yes.

*Question*: Concerning the left lung, which anatomical finding is involved, enlarged hilum or cyst/bullae?
*CoT answer*: Find the left lung <cot_bbox_8>. For the enlarged hilum, observe the left hilar structures <cot_bbox_18>, and there is no enlarged hilum in the left hilar structures <cot_bbox_18>. Therefore, there is no enlarged hilum in the left lung <cot_bbox_8>. For the cyst/bullae, observe the left upper lung zone <cot_bbox_9>, the left mid lung zone <cot_bbox_10>, the left lower lung zone <cot_bbox_11>, and the left lung <cot_bbox_8>. Considering all observations, there is cyst/bullae in the left lung <cot_bbox_8>. So the answer is cyst/bullae.

As shown in Fig. A5, we provide 8 examples from the AOR-VQA to facilitate a better understanding of our multimodal CoT data.

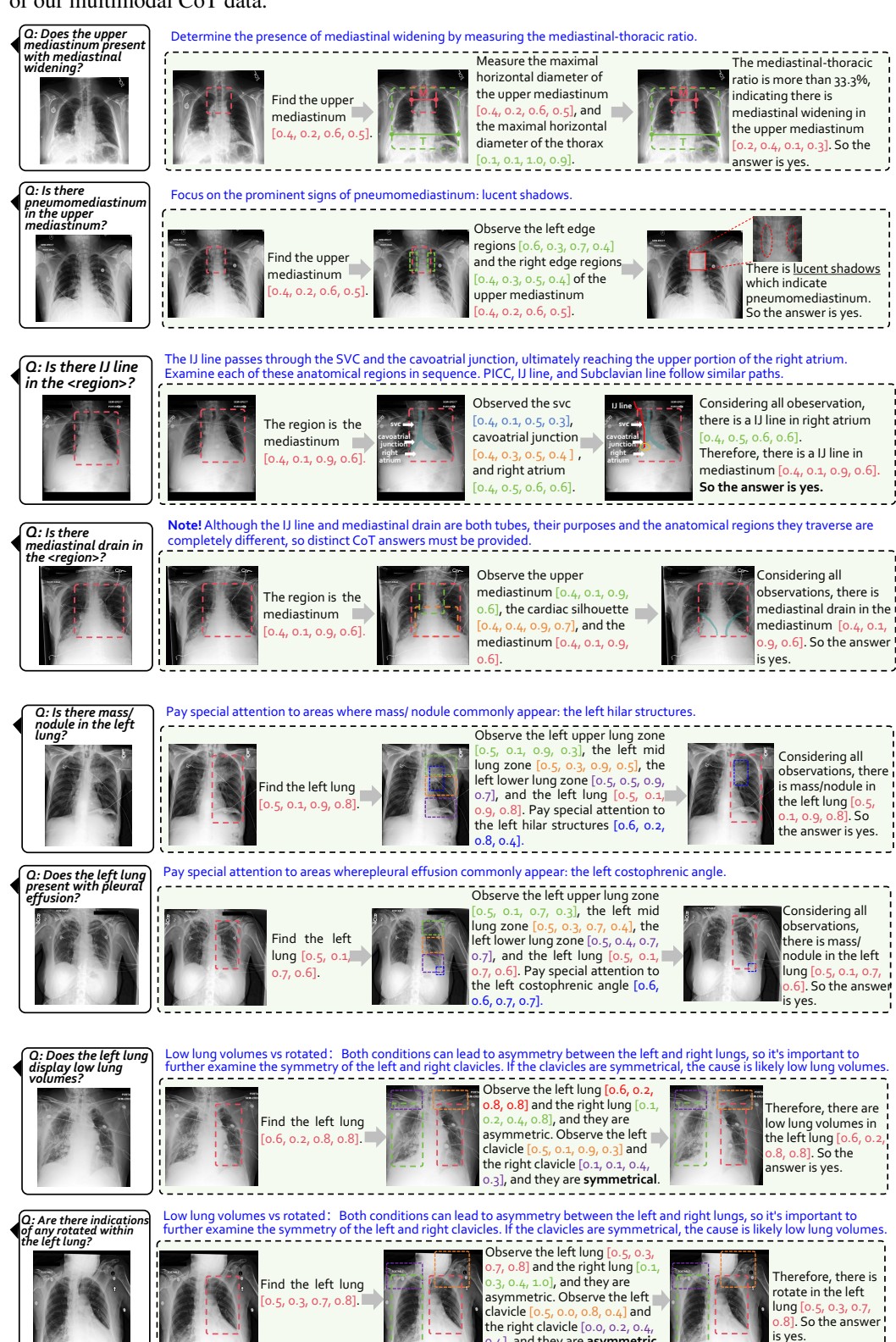

Figure A5: Samples from AOR-VQA.

### A.2.2 AOR-RG

For region report generation in AOR-RG, we adopt ASG's strict alignment approach and further optimize it. The details are as follows: First, for a given chest X-ray, we extract anatomical regions using the bounding box (bbox) coordinates from Chest ImaGenome Dataset, i.e., the objects listed in Table A2. Next, we utilize RadGraph to parse the corresponding paired reports. As shown in Table A9, the anatomical regions defined in RadGraph differ from those in Chest ImaGenome Dataset. To address this, we apply the alignment rules provided by ASG to ensure consistency. As illustrated in Fig. 3 (b) of the main manuscript, this alignment strategy guarantees a one-to-one correspondence between anatomical regions and sentences. This prevents scenarios where, for instance, merely detecting one side of the lung would lead to an incomplete statement like "The lung is..." without specifying the side.

Table A9: Objects in RadGraph.

| Objects in RadGraph | trachea, left hilar, right hilar, hilar unspec, left pleural, right pleural, pleural unspec, heart size, heart border, left diaphragm, right diaphragm, diaphragm unspec, retrocardiac, lower left lobe, upper left lobe, lower right lobe middle right lobe, upper right lobe, left lower lung, left mid lung, left upper lung left apical lung, left lung unspec, right lower lung, right mid lung, right upper lung right apical lung, right lung unspec, lung apices, lung bases, left costophrenic right costophrenic, costophrenic unspec, cardiophrenic sulcus, mediastinal, spine clavicle, rib, stomach, right atrium, right ventricle, aorta, svc, interstitium, parenchymal, cavoatrial junction, cardiopulmonary, pulmonary, lung volumes. |
|---|---|

Additionally, we further optimize the alignment method to handle cases where two different anatomical regions appear in the same short sentence, e.g., "Mediastinal and hilar contours are unremarkable." Using the previously mentioned method alone, this sentence would not be split to correspond separately to the mediastinal region and hilar region. To address this, we introduce a new splitting rule: if a short sentence describes two different anatomical regions simultaneously, it is split into two separate sentences. Thus, the original sentence is transformed into "Mediastinal is unremarkable." and "Hilar contours are unremarkable." Based on this, we further refine the alignment by distinguishing left/right and specific anatomical regions.

# B Experiment Details

## B.1 Prompt Details

We provide the detailed prompts for the three-stage training, along with their inputs and outputs. Here, [anatomical regions list] includes the objects listed in Table A2, and the responses highlighted in red indicate those for which loss needs to be calculated during training.

---

**Stage 1: Anatomical Region Recognition**

<image>In the conversation below, you are required to answer the category name based on what you see in the imagery inside a particular region. I will give you only one region each time. Categories containing {anatomical regions list}.

<region> `left lung`
<region> `mediastinum`

---

**Stage 2: Anatomical Region Grounding**

▶ **Task 1:** <image>In the conversation below, you are required to answer the category name based on the given region. The region is provided in coordinate form and imagery form. The coordinate form is $[x_1, y_1, x_2, y_2]$, with floating point number from 0 to 1. These values correspond to the top left $x$, top left $y$, bottom right $x$, and bottom right $y$. Categories containing {anatomical regions list}.

[0.50, 0.14, 0.89, 0.82] <region>`left lung`
[0.33, 0.28, 0.75, 0.80] <region>`mediastinum`

▶ **Task 2:** <image>In the conversation below, you are required to locate the corresponding region of the given category name in the image, and output its coordinates in the form of $[x_1, y_1, x_2, y_2]$, with floating point number from 0 to 1. These values correspond to the top left $x$, top left $y$, bottom right $x$, and bottom right $y$. I will give you only one category name each time.

left lung `[0.50, 0.14, 0.89, 0.82]`
mediastinum `[0.33, 0.28, 0.75, 0.80]`

---

**Stage 3: Instruction Tuning**

▶ **Task 1:** <image>provides an overview of the picture.

Question `cot answer`

▶ **Task 2:** <image>provides an overview of the picture. Please generate a radiology report based on this Chest X-ray.

`Full image report`

▶ **Task 3:** <image>provides an overview of the picture. Please generate a corresponding radiology report based on this region of the Chest X-ray.

<region>`region report sentence`

## B.2 Training Details

We supplement the hyperparameters for each stage in Fig. B10.

Table B10: Detailed training hyperparameters for AOR.

| Configuration | Stage 1 | Stage 2 | Stage 3 |
|---|---|---|---|
| Visual Encoder | CLIP ViT-L/14 | CLIP ViT-L/14 | CLIP ViT-L/14 |
| Model Init | LLaVA-1.5 | Stage 1 | Stage 2 |
| Global batch size | 128 | 128 | 128 |
| Learning rate | $2 \times 10^{-5}$ | $2 \times 10^{-5}$ | $2 \times 10^{-5}$ |
| Weight decay | 0 | 0 | 0 |
| Resolution | 336 | 336 | 336 |
| Training time | 6h | 9h | t-28h; r/t-43h |
| Epochs | 2 | 2 | 2 |
| Optimizer | AdamW | AdamW | AdamW |

**More Details About Region Encoder $\mathcal{R}$**   Following GPT4RoI [42], we introduce a region encoder $\mathcal{R}$ to provide fine-grained, multi-scale visual representations. The process includes:

(1) Feature Extraction: Four feature maps $z_j (j = 14, 17, 20, 23)$ are extracted from the image encoder and rescaled via bilinear interpolation to progressively larger scales $(\frac{1}{14}, \frac{2}{14}, \frac{4}{14}, \frac{8}{14})$, forming a multi-level feature pyramid $\{p_k\}_{k=1}^4$.

(2) Pyramid Shuffle Fusion: At each pyramid level $k$, features from adjacent levels are resized to match the resolution of level $k$, concatenated with the current feature map, and passed through convolutional layers to generate the fused representation $\hat{p}_k$.

$$\hat{p}_k = \text{Conv}\Big(\text{Concat}\big(\text{Resize}(p_{k-1}), \, p_k, \, \text{Resize}(p_{k+1})\big)\Big), \quad k = 1, 2, 3, 4.$$

(3) Region Feature Aggregation: Region features are extracted from $\{\hat{p}_k\}_{k=1}^4$ via RoIAlign, fused by convolutions, and pooled into a unified region representation for downstream reasoning tasks.

## B.3 Evaluation Details

**ROUGE-L**   ROUGE-L measures the length of the longest common subsequence (LCS) shared by the candidate and reference texts. It computes precision, recall, and $F_1$ based on the LCS length. We use the $F_1$ score of ROUGE-L to compare the generated radiology report with the ground truth report. For implementation, we use the `evaluate` library: https://github.com/huggingface/evaluate/tree/main/metrics/rouge.

**BERTScore**   BERTScore is a neural metric that leverages a pre-trained BERT model to evaluate the semantic similarity between text pairs. It computes pairwise cosine similarities between the contextualized token embeddings of the candidate and reference texts, then aggregates these similarities into precision, recall, and $F_1$ scores. In this work, we report the BERTScore $F_1$ measure to assess how closely the generated radiology reports match the ground truth reports. For implementation, we use the `evaluate` library: https://github.com/huggingface/evaluate/tree/main/metrics/bertscore.

**$F_1$CheXbert**   $F_1$CheXbert is computed using CheXbert, a Transformer-based model trained to identify 14 chest X-ray abnormalities from a radiology report. To evaluate the quality of generated reports, $F_1$CheXbert measures the $F_1$ score between CheXbert's predicted labels on the generated report and those on the ground truth report. Following previous work, the calculation focuses on five specific observations: atelectasis, cardiomegaly, consolidation, edema, and pleural effusion. For implementation, we use the `f1chexbert` library: https://github.com/jbdel/vilmedic.

## B.4 Result Details

### B.4.1 Intermediate Results of Stage 1

As shown in Table B11, after completing Stage 1 training, we conducted an initial evaluation of the model's anatomical region recognition capability. The average accuracy of anatomical region recognition is 98.37%, laying a foundation for subsequent training.

Table B11: Intermediate results of Stage 1.

| Anatomical Region | Acc | Anatomical Region | Acc |
|---|---|---|---|
| left costophrenic angle | 100.00 | right hemidiaphragm | 97.39 |
| right clavicle | 98.59 | upper mediastinum | 99.20 |
| trachea | 96.60 | left hilar structures | 98.80 |
| abdomen | 99.80 | svc | 98.80 |
| right hilar structures | 97.79 | left mid lung zone | 99.40 |
| carina | 99.60 | cardiac silhouette | 100.00 |
| right atrium | 99.00 | left upper lung zone | 97.58 |
| left lung | 99.60 | left apical zone | 96.98 |
| left hemidiaphragm | 93.60 | right lower lung zone | 97.20 |
| left lower lung zone | 95.20 | right lung | 100.00 |
| mediastinum | 99.60 | spine | 99.80 |
| left clavicle | 97.99 | right costophrenic angle | 100.00 |
| right apical zone | 95.79 | right upper lung zone | 99.00 |
| right mid lung zone | 98.80 | aortic arch | 98.80 |
| cavoatrial junction | 97.80 | | |

### B.4.2 Intermediate Results of Stage 2

Table B12 presents the intermediate results of Stage 2, validating the model's grounding capability. Additionally, using "[ ]" to represent coordinates achieved better results compared to the special token "<coor> </coor>".

Table B12: Intermediate results of Stage 2.

| format | R@0.3 | R@0.5 | R@0.7 |
|---|---|---|---|
| [ ] | **97.08** | **92.05** | **79.20** |
| <coor> | 97.05 | 91.88 | 78.37 |

### B.4.3 Results Details of Stage 3

Table B13, B14, B15 display the subclass results of AOR and other comparative methods on MIMIC-CXR-VQA, helping us analyze which tasks AOR excels in. As discussed in the main manuscript, AOR demonstrates a significant advantage in handling complex questions, such as choose and query types. By systematically analyzing multiple attributes involved in a question, the model is better equipped to perform logical reasoning and provide the correct answer. Additionally, for size-related questions, AOR improves answer accuracy by comparing the proportion of the heart or upper mediastinum relative to the entire thoracic cavity, leading to more precise evaluations.

Table B13: Detailed results of comparative experiments (verify).

| Method | Res | verify | | | | | |
|---|---|---|---|---|---|---|---|
| | | presence | abnormality | size | plane | gender | total |
| *General-domain LMM* | | | | | | | |
| LLaVA | $224^2$ | 75.91 | 77.02 | 75.32 | 95.45 | 50.76 | 75.97 |
| LLaVA-1.5 | $336^2$ | 74.66 | 74.96 | 73.62 | 96.97 | 85.61 | 75.25 |
| GPT4RoI | $224^2$ | 76.02 | 77.59 | 77.31 | 95.45 | **93.18** | 77.16 |
| VoCoT | $448^2$ | 76.48 | 76.95 | 73.62 | 70.45 | 76.52 | 76.17 |
| *Medical-domain LMM* | | | | | | | |
| LLaVA-Med | $224^2$ | 75.32 | 75.53 | 74.47 | 94.70 | 78.79 | 75.71 |
| Med-Flamingo | $224^2$ | - | - | - | - | - | - |
| XrayGPT | $224^2$ | 61.21 | 57.94 | 58.87 | 53.03 | 53.03 | 60.00 |
| CheXagent | $448^2$ | 76.65 | 76.31 | 70.07 | 50.00 | 56.06 | 75.02 |
| MedVLM-R1 | *Dyn* | - | - | - | - | - | - |
| AOR(Ours)-t | $336^2$ | **79.48** | **78.51** | **84.97** | **99.24** | **93.18** | **80.48** |
| AOR(Ours)-r/t | $336^2$ | 79.02 | 79.01 | 93.76 | 98.49 | 93.94 | 81.17 |

Table B14: Detailed results of comparative experiments (choose).

| Method | Res | choose | | | | | |
|---|---|---|---|---|---|---|---|
| | | attribute | abnormality | anatomy | plane | gender | total |
| *General-domain LMM* | | | | | | | |
| LLaVA | $224^2$ | 52.72 | 51.63 | 49.66 | 93.18 | 60.61 | 56.07 |
| LLaVA-1.5 | $336^2$ | 50.64 | 53.12 | 51.34 | 95.45 | 90.91 | 58.70 |
| GPT4RoI | $224^2$ | 53.69 | 40.06 | 46.31 | 96.21 | **94.70** | 56.47 |
| VoCoT | $448^2$ | 43.91 | 37.39 | 36.91 | 93.18 | 83.33 | 48.79 |
| *Medical-domain LMM* | | | | | | | |
| LLaVA-Med | $224^2$ | 50.32 | 51.04 | 52.01 | 95.45 | 91.67 | 58.31 |
| Med-Flamingo | $224^2$ | - | - | - | - | - | - |
| XrayGPT | $224^2$ | 29.97 | 40.95 | 43.62 | 78.79 | 49.24 | 40.97 |
| CheXagent | $448^2$ | 27.72 | 24.33 | 21.81 | 91.67 | 52.27 | 33.49 |
| MedVLM-R1 | *Dyn* | - | - | - | - | - | - |
| AOR(Ours)-t | $336^2$ | **66.51** | **69.44** | **64.43** | **98.49** | **94.70** | **71.96** |
| AOR(Ours)-r/t | $336^2$ | 63.62 | 69.73 | 60.40 | 98.48 | 96.21 | 70.19 |

Table B15: Detailed results of comparative experiments (query).

| Method | Res | query | | | | | |
|---|---|---|---|---|---|---|---|
| | | attribute | abnormality | anatomy | plane | gender | total |
| *General-domain LMM* | | | | | | | |
| LLaVA | $224^2$ | 60.70 | 56.20 | 64.79 | 86.07 | 55.30 | 58.87 |
| LLaVA-1.5 | $336^2$ | 60.79 | 53.77 | 62.43 | 96.72 | 87.12 | 56.10 |
| GPT4RoI | $224^2$ | 60.40 | 57.14 | 67.32 | **98.36** | 90.15 | 60.54 |
| VoCoT | $448^2$ | 61.22 | 55.38 | 62.80 | 97.54 | 81.82 | 60.70 |
| *Medical-domain LMM* | | | | | | | |
| LLaVA-Med | $224^2$ | 61.48 | 57.19 | 65.98 | 97.54 | 90.15 | 60.37 |
| Med-Flamingo | $224^2$ | - | - | - | - | - | - |
| XrayGPT | $224^2$ | 44.06 | 34.18 | 37.77 | 23.50 | 50.00 | 24.07 |
| CheXagent | $448^2$ | 37.42 | 34.09 | 46.55 | 68.85 | 28.03 | 48.49 |
| MedVLM-R1 | *Dyn* | - | - | - | - | - | - |
| AOR(Ours)-t | $336^2$ | **66.71** | **62.69** | **69.43** | **98.36** | **91.67** | **65.05** |
| AOR(Ours)-r/t | $336^2$ | 66.95 | 62.75 | 71.18 | 100 | 93.18 | 65.49 |

## B.5 Scalability of CoT Construction

Using VQA-RAD as an example, we generate CoT answers for each sample as follows: **Keyword Mapping** We use LLMs (GPT-4o) to extract relevant terms from each question and map them to our predefined 68 attributes, 5 categories, and 38 objects. Unmapped terms are reviewed manually (though rare, as explained later). **Sample Expansion** Once mapped, we generate CoT answers based on our 2,812 predefined CoT types. Key findings:

- **Performance Gain** Defining 2,812 CoT types enables LLMs to generate CoT answers for any CXR dataset. The LLMs can easily map attributes/categories and objects (e.g., the base of the right→lower right lung) and proceed to construct CoT answer. As shown in Table 5, incorporating CoT answers improves performance by 3.60% after 1 epoch and 5.97% after 3 epochs of fine-tuning, demonstrating the generalizability of our approach to new datasets.

- **Data Quality Enhancement** We examine failure cases in mapping and identify two main points: (a) Non-specific anatomical regions, e.g., "the lungs" cover both sides. We resolve this by splitting the region into left and right sides for individual analysis and then aggregating the results. (b) Unmapped attributes, i.e., cardiomyopathy and aortopulmonary window, could not be mapped. Radiologist consultation confirmed these are challenging to assess via a single X-ray, as heart size and shape alone are insufficient. Similarly, the aortopulmonary window overlaps with adjacent structures on chest X-rays. These findings validate the accuracy and clinical reliability of our 68 predefined attributes and help refine low-quality or non-specific samples in existing datasets.

# C  Case Study

## C.1  Case Study: Visual Question Answering (VQA) on MIMIC-CXR-VQA

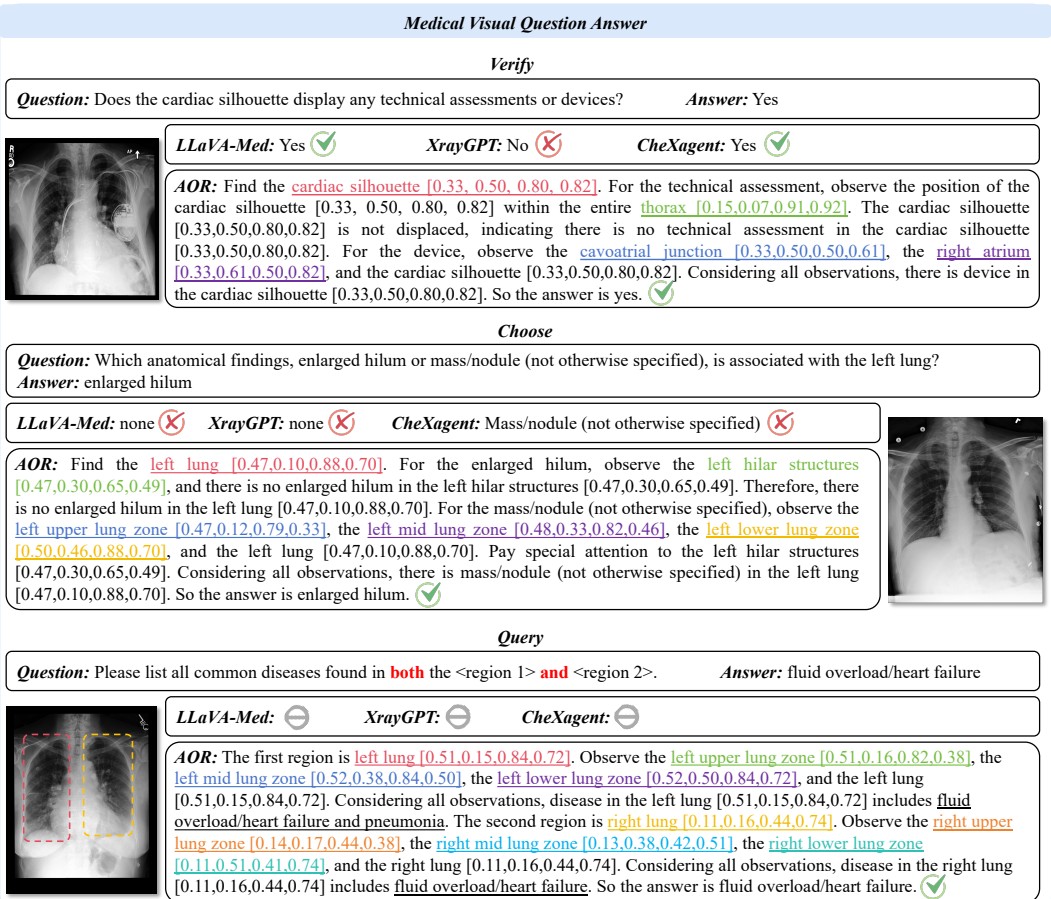

Figure C6: Cases of VQA on MIMIC-CXR-VQA: Leveraging region-level understanding and multi-step reasoning, AOR achieves superior performance.

## C.2 Case Study: Visual Question Answering (VQA) on VQA-RAD and CheXpert

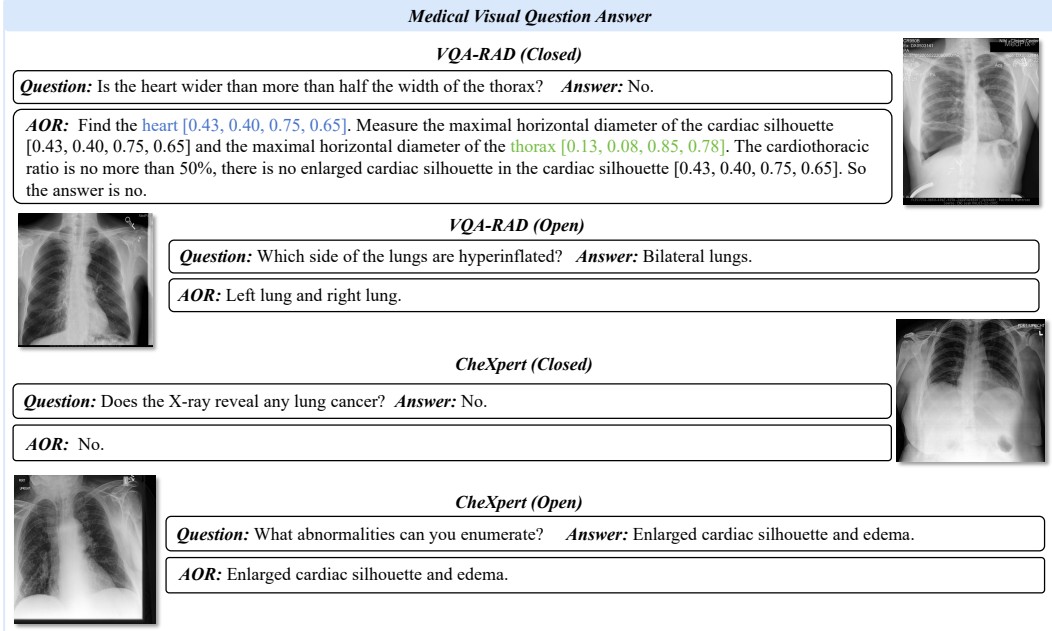

Figure C7: Cases of VQA on VQA-RAD and CheXpert: For local-level questions, even on unseen VQA-RAD data, AOR can perform reasoning and provide correct answers. For global-level questions, AOR is also capable of synthesizing information to make a final judgment.

## C.3 Case Study: Full-Image Report Generation on MIMIC-CXR

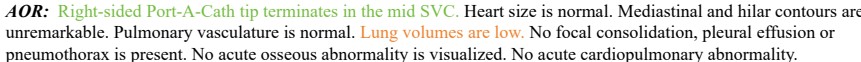

*Full Image Report Generation*

> ***GroundTruth:*** AP upright and lateral views of the chest were provided. A Port-A-Cath resides over the right chest wall with catheter tip in the expected region of the mid SVC. Lung volumes are low. No effusion or pneumothorax. No convincing evidence for pneumonia or CHF. The cardiomediastinal silhouette is normal. Bony structures are intact. No free air below the right hemidiaphragm. No acute findings.

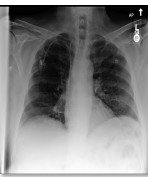

> ***AOR:*** Right-sided Port-A-Cath tip terminates in the mid SVC. Heart size is normal. Mediastinal and hilar contours are unremarkable. Pulmonary vasculature is normal. Lung volumes are low. No focal consolidation, pleural effusion or pneumothorax is present. No acute osseous abnormality is visualized. No acute cardiopulmonary abnormality.

> ***LLaVA-Med:*** The chest X-ray appears to be normal, which means that there are no visible signs of abnormalities or issues in the lungs, heart, or surrounding structures. This is a positive finding, as it suggests that the patient's
>
> ***Med-flamingo:*** This is a frontal chest radiograph of a 55-year-old male patient with a history of a cough and fever. The patient is in a supine position. The cardiac silhouette is normal. The lungs are clear. The trachea is central. The hila are normal. The lung fields are clear. The mediastinum is normal. The ribs are normal. The bony thorax is normal.
>
> ***XrayGPT:*** The chest x-ray shows a normal heart size, unremarkable mediastinal and hilar contours, and clear lungs without any focal consolidation. There is no pleural effusion or pneumothorax. The impression is that there is no acute cardiopulmonary process.
>
> ***CheXagent:*** In comparison with the study of ___, there is little change and no evidence of acute cardiopulmonary disease. No pneumonia, vascular congestion, or pleural effusion. Central catheter remains in place.

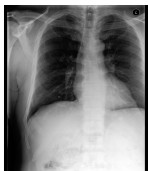

> ***GroundTruth:*** Heart size is normal. The mediastinal and hilar contours are normal. The pulmonary vasculature is normal. Lungs are clear. No pleural effusion or pneumothorax is seen. There are no acute osseous abnormalities. Moderate degenerative changes are demonstrated within the imaged thoracic spine. No acute cardiopulmonary abnormality.

> ***AOR:*** The lungs are clear. The cardiomediastinal silhouette is within normal limits. No acute osseous abnormalities. No acute cardiopulmonary process.

> ***LLaVA-Med:*** The chest X-ray appears to be normal, which means that there are no visible signs of abnormalities or issues in the lungs, heart, or surrounding structures.
>
> ***Med-flamingo:*** This is a frontal chest radiograph of a 55-year-old male patient with a history of a cough and fever. The patient is in a supine position. The cardiac silhouette is normal. The lungs are clear. The trachea is central. The hila are normal. The lung fields are clear. The mediastinum is normal. The bones are normal.
>
> ***XrayGPT:*** The chest x-ray shows clear lungs with no signs of consolidation, effusion, or edema. The cardiomediastinal silhouette is normal, and there are no acute osseous abnormalities. The impression is that there is no acute cardiopulmonary process.
>
> ***CheXagent:*** Heart size is normal. The mediastinal and hilar contours are normal. The pulmonary vasculature is normal. Lungs are clear. No pleural effusion or pneumothorax is seen.

Figure C8: Cases of full-image report generation on MIMIC-CXR: AOR demonstrates strong detail-capturing capabilities.

## C.4  Case Study: Region-Level Report Generation on MIMIC-CXR

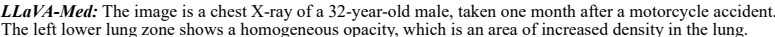

***Region Report Generation***

*Please generate a corresponding radiology report based on left lower lung zone of the Chest X-ray*
*GroundTruth:* There is mild left base atelectasis.

*AOR:*  There is mild left basilar atelectasis.

*LLaVA-Med:* The image is a chest X-ray of a 32-year-old male, taken one month after a motorcycle accident. The left lower lung zone shows a homogeneous opacity, which is an area of increased density in the lung.

*Med-flamingo:* The left lower zone is a normal zone. The left upper zone is a normal zone. The left middle zone is a normal zone.

*XrayGPT:* The chest x-ray shows a normal heart size, with normal mediastinal and hilar contours. The pulmonary vasculature is also normal, and the lungs are clear. There are no signs of pleural effusion or pneumothorax. The impression is that there is no acute cardiopulmonary abnormality.

*CheXagent:* In comparison with the study of ___, there is little change and no evidence of acute cardiopulmonary disease. No pneumonia, vascular congestion, or pleural effusion.

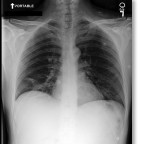

*Please generate a corresponding radiology report based on svc of the Chest X-ray*
*GroundTruth:* The picc is now in the mid svc. The picc has been pulled back and is now within the mid svc.

*AOR:* Right picc line tip is at the level of mid svc.

*LLaVA-Med:* The chest X-ray appears to be normal, which means that there are no visible signs of abnormalities or issues in the lungs, heart, or surrounding structures.

*Med-flamingo:* The patient is a 60-year-old male with a history of hypertension and diabetes. He is currently being treated for a right lower lobe pneumonia. The patient is currently on a beta-blocker for his hypertension.

*XrayGPT:* The chest x-ray shows a normal heart size, normal mediastinal and hilar contours, and clear lungs. There is no evidence of pleural effusion or pneumothorax. The impression is that there is no acute cardiopulmonary process.

*CheXagent:* In comparison with the study of ___, the patient has taken a much better inspiration. There is no evidence of acute pneumonia, vascular congestion, or pleural effusion.

Figure C9: Cases of region report generation on MIMIC-CXR: AOR can generate accurate, grounded captions.

# D    Limitations and Future Work

**Frontal and Lateral Images**    AOR only utilizes frontal images for experiments, while lateral images also play a significant role in clinical practice. However, it is worth noting that the use of lateral images is far more complex than simply encoding and concatenating them with frontal images. Lateral images inherently provide rich CoT reasoning. For example, as shown in Fig. C8, when only the frontal image is provided as input, all medical LMMs fail to detect spine degenerative change, which further validates the importance of lateral images in certain cases. Similarly, when pulmonary abnormalities obscure the hilum in frontal images, lateral images are essential for further observation and confirmation. Leveraging lateral images will be an important direction for our future work.

**Temporal Images**    Currently, AOR only analyzes static CXRs. However, in real-world scenarios, diseases progress over time, and CXRs are often presented as a sequence captured at different time points. Therefore, the ability to understand temporal images is also crucial, which we hope to explore further in future work.

**Extension to More Modalities**    AOR focuses on CXR interpretation, providing anatomical ontology-guided reasoning capabilities. Based on this, we discuss the potential of extending AOR to additional modalities: (1) Extension to modalities with clearly defined anatomical regions. For modalities such as chest CT, abdominal CT, and brain MRI, which have clearly defined anatomical regions, we can use these existing divisions as anchors to construct corresponding CoT answers, thereby mimicking radiologists' reasoning processes in clinical decision-making. (2) Extension to modalities without explicitly defined anatomical regions. We also explore structures such as bones or teeth, which do not have conventional anatomical zoning. We consulted certified radiologists to design reasoning flows based on their unique structural features. For example, in bone structures, reasoning can be organized around specific vertebrae or ribs; in tooth structures, it can be decomposed into crown, root, jaw, condyle, airway, etc.

# E    Impact Statement

For our research, we utilize the source dataset under the PhysioNet license, ensuring compliance with the required credentials and permissions, and eliminating the risk of privacy infringement. By empowering Medical LMMs with anatomy-centric reasoning capabilities, we offer a new paradigm for interactive and explainable LMMs in medical imaging analysis. Experiments demonstrate AOR's superior performance in both VQA and report generation tasks, revealing its potential in supporting clinical decision-making.

