# OpenReview forum: "AOR: Anatomical Ontology-Guided Reasoning for Medical Large Multimodal Model in Chest X-Ray Interpretation"
_NeurIPS.cc/2025/Conference — NeurIPS 2025 poster_

### Official Review · Reviewer_wKs5 · 2025-06-05

**Clarity:** 3
**Significance:** 3
**Originality:** 3
**Rating:** 5
**Confidence:** 4

**Summary:**

This paper presents a framework called AOR (Anatomical Ontology-Guided Reasoning), which aims to improve the reasoning and interpretability of large multimodal models for chest X-ray analysis. The authors focus on two key limitations of current models: the lack of region-level understanding and the reliance on single-step prediction. To tackle this, they propose a three-stage training approach that includes anatomical region recognition, grounding, and instruction tuning. A major component of their work is the construction of AOR-Instruction, a large-scale dataset developed under expert guidance, which incorporates structured reasoning based on anatomical ontologies. The method supports both textual and optional visual prompts, and is evaluated on several tasks including VQA and report generation. Experimental results show clear improvements over existing medical vision-language models, especially in cases requiring step-by-step reasoning or regional grounding.

**Questions:**

1. It would be helpful if the authors could clarify whether any manual review or quality check was conducted on the generated AOR-VQA or AOR-RG samples, especially given the scale of the dataset. Additionally, is there any plan to release the dataset (or a portion of it) to facilitate future research?
2. Since the backbone of the proposed model is a general-purpose LLM, one would expect it to retain some degree of open-domain or out-of-distribution (OOD) reasoning capability. I wonder if the authors have considered evaluating the model on more global or metadata-related questions (e.g., patient age, gender, image view) that are present in MIMIC-CXR-VQA. Such evaluation could help assess whether the model preserves generalization ability beyond the region-level tasks it was primarily trained for.
3. In Table 3, it is interesting to observe that the setting using CoT alone (without coordinate or region features) already achieves the best performance on the “verify” question type, nearly matching or slightly outperforming the full AOR setting. Could the authors comment on why CoT alone is particularly effective for this category, and whether this suggests that “verify” questions benefit more from structured textual reasoning than from spatial or visual cues?

**Ethical Concerns:**

["NO or VERY MINOR ethics concerns only"]

**Final Justification:**

After reading the authors' rebuttal, most of my concern have been resoloved. I will keep my positive score.

**Limitations:**

While the paper addresses some limitations, one important concern is not discussed. The proposed use of region-level reasoning is arguably more suited to 3D imaging modalities, where anatomical regions are better defined and spatially consistent. In 2D chest X-rays, the projection inherently compresses anatomical structures (e.g., the heart), and region definitions may not fully capture the relevant semantic or clinical context. As a result, some region-level supervision might be misaligned with true anatomy, and the benefit of region guidance may be overestimated in this setting. It would be helpful for the authors to reflect on this limitation and discuss whether their approach could generalize or even perform better in volumetric imaging tasks like CT or MRI.

**Paper Formatting Concerns:**

No major formatting issues noticed. The paper adheres to the NeurIPS 2025 formatting guidelines, including page limit, anonymity, and layout.

**Quality:**

3

**Strengths And Weaknesses:**

Strengths

1. The paper proposes a structured and well-motivated approach that introduces region-level information in a staged manner, which aligns with how radiologists interpret medical images.

2. The authors construct a large instruction dataset based on anatomical ontologies and Chain-of-Thought reasoning, which required domain expertise and could benefit the broader medical AI community.

3. Experiments are thorough and cover both visual question answering and report generation tasks, with consistent improvements over strong baselines.

4. The method enhances interpretability through region-aware reasoning, which is especially valuable in high-stakes domains like healthcare.

Weakness

1. While the model is built upon a general-purpose LLM, the paper does not explore its generalization ability to out-of-distribution question types such as age or gender, which are not relevant and present in the dataset.

2. The ablation study in Table 3 shows that CoT alone achieves nearly the best result on “verify” questions, but the paper lacks discussion on why coordinate and region features seem to offer limited additional benefit in this case.

---

> ### Author Rebuttal · Authors · 2025-07-31
>
> Thank you for recognizing our work and the valuable comments. We hope that the following point-by-point responses address your concerns.
>
> ### **Q1: Quality Control and Release Plan for AOR-VQA and AOR-RG**.
>
> Yes, we performed manual review for quality control during the dataset construction process. Details for each dataset are provided below:
>
> **AOR-VQA**: From ontology design to CoT construction, the entire process was guided by three expert physicians. We constructed **2,812 CoT types**, derived from combinations of question levels, anatomical regions, and findings. Each CoT type was thoroughly reviewed by at least two physicians, with ~1,900 types reviewed per physician. Disagreements were resolved through collective discussion.
>   These reviewed CoT types were then expanded to individual QA samples through logical composition. Since this step involves deterministic transformations, it theoretically introduces no additional risk of error. To further ensure data quality, we randomly sampled **1% of the ~290,000 QA pairs post-generation for manual review.** These samples were evenly distributed among three physicians.
>
> **AOR-RG**: We constructed AOR-RG based on the rules proposed in ASG [1], which stated that their anatomical region–sentence alignment paradigm was developed under the guidance of radiologists. Building on this foundation, we optimized the alignment method to handle cases where two different anatomical regions appear in the same short sentence, introducing new rules to split such sentences into two separate ones. Similarly, we randomly sampled **0.5% of the ~399,000 region–sentence pairs for manual review** by three physicians to ensure that no recurring or quality-compromising errors were present.
>
> **Dataset Release Plan**: We will publicly release the AOR-VQA and AOR-RG datasets and are in the process of preparing the files and documentation for upload to PhysioNet.
>
> ### **W1 & Q2: Generalization to Global and Out-of-Distribution Questions**
>
> **Our Design Choice: Focus on Higher-Priority CXR Tasks.** Our study was intentionally designed to focus on clinically relevant, fine-grained reasoning tasks that are of high priority in real-world radiology, such as localizing lesions, interpreting complex findings, and generating detailed report sections. In contrast, we assigned a lower clinical priority to meta-level information (e.g., patient age or gender) that can be readily obtained from hospital systems, and thus did not incorporate detailed reasoning chains for them in our primary task design. This focus allows our model, AOR, to excel at its core purpose.
>
> **Performance on Global Questions in AOR-QA.** Despite this focus on fine-grained tasks, our model is still highly capable of handling the global or metadata-related questions present in our dataset. AOR-QA includes all question types from MIMIC-CXR-VQA, and as shown in Fig. A1 (Appendix), 'Plane' (image view) and 'Gender' questions are indeed part of it. Our model achieves accuracies of ~95% on these question types (Tables B13-B15), demonstrating strong performance on the non-regional tasks available in the dataset.
>
> **Further Evaluation on OOD Reasoning.** To more directly address the important point about out-of-distribution (OOD) generalization, we conducted further tests. We evaluated AOR on a subset of VQA-RAD questions, which are entirely OOD with respect to our training data (e.g., knowledge-based questions unrelated to the image). AOR achieves 42.86% accuracy on closed questions and 10.00% on open ones. While these results are preliminary, they provide a valuable benchmark.
>
> **Future Research Direction.** These findings reinforce our view that extending the model’s capabilities to open-domain and OOD tasks is a valuable next step. As this work lays the foundation for anatomy-oriented reasoning, we see a clear path forward. We believe that using stronger base models (e.g., Qwen2.5-VL) and leveraging retrieval-augmented techniques are promising directions to improve AOR on these tasks. We particularly aim to explore efficient learning from small samples, a critical challenge in the medical domain.
>
> ### **W2 & Q3: Further discussion on the Comparison of different CoT representations.**
>
> **Further discussion**: Among the three types of questions, verify questions are relatively easier. The binary nature of verify questions (i.e., yes/no) compresses the model’s output space, making it more difficult to demonstrate performance improvements once a high accuracy level is reached (e.g., our baseline ID 2 achieves 80.69% on verify). In contrast, choose and query questions impose higher demands on the model’s reasoning correctness. Compared to ID 2 (CoT-only), ID 4 (full AOR) shows performance gains of **+2.64%** and **+2.03%** on choose and query questions respectively, highlighting the importance of spatial and visual cues.
>
> **Additional zero-shot evaluation on VQA-RAD**: To further support this observation, we conducted an ablation study on the VQA-RAD dataset under a zero-shot setting. As shown in Table 1, incorporating spatial and visual cues yields improvements of **5.48%** on closed-form (verify-like) questions. This demonstrates that such cues enhance the model’s comprehension and reasoning abilities on previously unseen data distributions.
>
> Table 1. Comparison of CoT representations on VQA-RAD.
> | Exp_ID |  coor | region |  CoT  |   closed  |    open   |
> |:------:|:-----:|:------:|:-----:|:---------:|:---------:|
> |    1   | **✗** |  **✗** | **✗** |   47.74   |   16.90   |
> |    2   | **✗** |  **✗** | **✓** |   52.05   |   19.20   |
> |    3   | **✓** |  **✗** | **✓** |   54.50   |   21.91   |
> |    4   | **✓** |  **✓** | **✓** | **57.53** | **24.99** |
>
> ### **Limitations:  generalization to 3D imaging modalities.**
>
> **Effectiveness on 2D chest X-rays**: We acknowledge that 2D chest X-rays are subject to certain degrees of projection-induced compression. However, chest X-rays remain the first-line imaging tool  in routine clinical practice due to their low cost, minimal radiation exposure, and broad accessibility. Our anatomical region definitions are based on widely used clinical zoning standards and have been reviewed and validated by experienced radiologists, ensuring the **semantic and diagnostic utility** of the defined regions . Therefore, despite the limitations of  projection, region-level information in 2D chest X-rays still holds substantial clinical value and research relevance.
>
> **Extension to 3D chest CT**: We fully agree that 3D imaging modalities offer richer anatomical information and provide a more consistent spatial context. Through our practical experience, we found this to be a highly meaningful yet very challenging task. For example, in 2D chest X-rays, dividing the heart into two or three subregions is usually sufficient to support most reasoning needs. However, chest CT, by its nature, contains much richer anatomical information. The heart in CT images can be subdivided into eight distinct sub-anatomical regions, e.g., the left and right atria, ventricles, myocardium, and so on. Extending AOR  to such modalities is an important direction of our ongoing work.
>
> To this end, we have curated **a 3D chest CT dataset with over 130 anatomical regions**, compared to 29 in the 2D X-ray setting. The dataset includes 2,070 CT scans, split 80/20 into training and test sets.We replaced the 2D image encoder with a 3D-compatible variant, and evaluated region-level report generation.
>
> As shown in Table 2, our method **achieves an average performance improvement of 8.66%** over baselines lacking anatomical decomposition, even with limited data. These results affirm the potential of the AOR framework to generalize effectively to volumetric modalities.
>
> Table 2. Performance of CT region report generation.
> | ID                  | R-L   | BERTScore | F1CheXbert |
> |------------------------------|:-------:|:-----------:|:------------:|
> | 1 | 8.09  | 68.33     | 13.94      |
> | 2   | **16.45** | **79.18**     | **20.72**      |
>
> **Extension to other more modalities**: Beyond chest CT, we are exploring generalization to other modalities and regions within the oral cavity. In cases like dental imaging, where conventional anatomical zoning is not well established, we collaborate with radiologists to design structure-aware reasoning flows by incorporating clinically important components commonly referenced in dental practice, such as crowns, roots, jaws, mandibular canals, and alveolar processes.
>
> This broader discussion will be added under the Limitations and Future Work section in the revised manuscript.
>
> ---
> [1] Li Q, Yan X, Xu J, et al. *Anatomical structure-guided medical vision-language pre-training.* MICCAI 2024.

---

> > ### Comment · Reviewer_wKs5 · 2025-08-09
> >
> > Thank you for providing the additional evidence. My concerns have been addressed, and I will keep my current score.

---

### Official Review · Reviewer_SRND · 2025-06-24

**Clarity:** 2
**Significance:** 2
**Originality:** 3
**Rating:** 4
**Confidence:** 4

**Summary:**

This paper introduces the Anatomical Ontology-Guided Reasoning (AOR) framework, which leverages multi-modal region-level anatomical correlation for step-by-step reasoning. Two datasets with multi-modal CoT answers are constructed to train AOR for both question-answering and report-generation. The proposed AOR shows strong results across several benchmarks.

**Questions:**

1. Could other models except the AOR model, such as LLava-med, gain performance enhancement by supervised fine-tuning with CoT data from these datasets?

2. Is it necessary to employ an independent encoder to extract local information? Additionally, the author should offer more details about the region encoder.

3. Will the trained AOR model generate inaccurate reasoning steps, such as incorrect identification of anatomical regions and their attributes?

**Ethical Concerns:**

["NO or VERY MINOR ethics concerns only"]

**Final Justification:**

After reading the authors' rebuttal, most of my concern have been resoloved. I will keep my positive score.

**Limitations:**

The proposed method only focuses on Chest X-Ray; the generalization capability across different tissues and imaging modalities of the proposed data generation pipeline and AOR architecture is unknown.

**Paper Formatting Concerns:**

None.

**Quality:**

3

**Strengths And Weaknesses:**

**Strengths:**
- This paper proposes a new medical dataset with CoT reasoning via exploring objects' relationships. For medical VLLMs, it is hard to generate CoT reasoning by distilling a stronger reasoning model, as there are no models that are strong enough to consistently provide reliable reasoning.
- The proposed data generation pipeline is sound and effective. Performing step-by-step reasoning by analyzing the correlation across anatomical regions facilitates the generation of more interpretable answers for medical QA and report generation tasks.
- The paper is well-written; the author provided a detailed description of the proposed methodology. Extensive experiments are conducted to demonstrate the effectiveness of the AOR model.


**Weaknesses:**
- Lacks evaluating how the newly introduced AOR-QA and AOR-RG datasets impact the performance of other medical multi-modal models.
- Lacks the ablation on the region encoder; it is unclear whether the performance improvement in the AOR model is caused by higher data quality or a specifically designed model architecture.

---

> ### Author Rebuttal · Authors · 2025-07-31
>
> Thank you for your recognition and valuable suggestions. We hope the following point-by-point responses address your concerns.
> ### **W1 & Q1: Effectiveness of AOR-VQA and AOR-RG in enhancing other LMMs.**
> To evaluate the impact of our proposed  AOR-VQA and AOR-RG datasets on other multimodal models, we fine-tuned **LLaVA-Med** and **GPT4RoI** using these datasets.
>
> As shown in Table 1 and Table 2:
> - **ID-1** refers to models trained on MIMIC-CXR-VQA (w/o CoT) and MIMIC-CXR (report generation with full images only).
> - **ID-2** denotes models fine-tuned with  our AOR-VQA dataset (w/ CoT) and AOR-RG (including both full-image and region-level report generation).
>
> The results are as follows:
> - **LLaVA-Med** achieves a +3.20% improvement on VQA and +5.81% on report generation.
> - **GPT4RoI** gains +4.62% and +6.69%, respectively.
>
> These results indicate the effectiveness of AOR-VQA and AOR-RG for other LMMs.
>
> Table 1. Performance on the VQA task.
> | Model     | Exp_ID |              | MIMIC-CXR-VQA |              |   |   | VQA-RAD      |              |   |   | CheXpert     |              |
> |-----------|:--------:|--------------|---------------|--------------|---|---|--------------|--------------|---|---|--------------|--------------|
> |           |        | verify       | choose        | query        |   |   | closed       | open         |   |   | closed       | open         |
> | LLaVA-Med | 1      | 76.34        | 57.58         | 60.76        |   |   | 48.09        | 23.05        |   |   | 58.02        | 39.50        |
> |           | 2      | **77.57**(+1.23) | **66.97**(+0.39)  | **62.17**(+1.41) |   |   | **50.56**(+1.47) | **26.22**(+3.17) |   |   | **60.79**(+2.77) | **42.48**(+2.98) |
> | GPT4ROI   | 1      | 76.17        | 56.34         | 60.39        |   |   | 45.59        | 16.49        |   |   | 54.55        | 35.66        |
> |           | 2      | **78.37**(+2.20)x | **69.14**(+12.8)  | **62.99**(+2.60) |   |   | **49.63**(+4.04) | **19.94**(+3.45) |   |   | **59.41**(+4.86) | **38.07**(+2.41) |
>
> Table 2. Performance on full image report generation.
> | Method    | Exp_ID | R-L          |   |   | BERTScore    |   |   | F1CheXbert   |
> |-----------|:------:|--------------|---|---|--------------|---|---|--------------|
> | LLaVA-Med |    1   | 23.68        |   |   | 83.13        |   |   | 37.60        |
> |           |    2   | **25.89**(+2.21) |   |   | **83.29**(+0.16) |   |   | **40.51**(+2.91) |
> | GPT4ROI   |    1   | 23.76        |   |   | 83.23        |   |   | 39.45        |
> |           |    2   | **25.13**(+1.37) |   |   | **83.86**(+0.63) |   |   | **44.11**(+4.66) |
>
> Table 3. Performance on region report generation.
> | Method    | Exp_ID | R-L           |   |   | BERTScore    |   |   | F1CheXbert    |
> |-----------|:------:|---------------|---|---|--------------|---|---|---------------|
> | LLaVA-Med |    1   | 17.24         |   |   | 80.76        |   |   | 24.07         |
> |           |    2   | **34.26**(+17.02) |   |   | **82.76**(+2.00) |   |   | **34.62**(+10.55) |
> | GPT4ROI   |    1   | 13.50         |   |   | 78.58        |   |   | 26.66         |
> |           |    2   | **33.82**(+20.32) |   |   | **82.96**(+4.38) |   |   | **35.42**(+8.78)  |/
>
> ### **W2 & Q2: More details and ablation study on the region encoder.**
> **More details.** Our region encoder is designed to provide fine-grained, multi-scale visual representations. The process includes:
>
> 1. **Feature Extraction**: Four feature maps $z_j(j = 14, 17, 20, 23)$ are extracted from the image encoder and rescaled via bilinear interpolation to progressively larger scales $(\frac{1}{14}, \frac{2}{14}, \frac{4}{14}, \frac{8}{14})$, forming a multi-level feature pyramid {$p_k$}$_{k=1}^4$.
> 2. **Pyramid Shuffle Fusion**: At each pyramid level $k$, features from adjacent levels are resized to match the resolution of level $k$, concatenated with the current feature map, and passed through convolutional layers to generate the fused representation $\hat{p}_k$.
> $$
> \hat{p}_k=\mathrm{Conv}(\mathrm{Concat}(\mathrm{Resize}(p\_{k-1}), \ p\_k ,\ \mathrm{Resize}(p\_{k+1}))), \ k = 1, 2, 3, 4
> $$
> 3. **Region Feature Aggregation**: Region features are extracted from {$\hat p_k$}$_{k=1}^4$ via RoIAlign, fused by convolutions, and pooled into a unified region representation for downstream reasoning tasks.
>
>
> **Ablation study.** As shown in Table 3 (on MIMIC-CXR-VQA), removing the region encoder (Only RoIAlign) results in a performance drop of 1.06%. Simply cropping region features using RoIAlign from a frozen image encoder leads to inaccurate region embeddings, highlighting the advantage of our multi-scale, trainable region encoder. In future work, we plan to explore more efficient image and region representations to further amplify the benefits brought by fine-grained visual cues.
>
> Table 3. Region encoder ablation on MIMIC-CXR-VQA.
> | Condition          |        |   VQA  |       |   |   | Referring |   |   |       | Grounding |       |
> |--------------------|:------:|:------:|:-----:|---|---|-----------|---|---|:-----:|:---------:|:-----:|
> |                    | verify | choose | query |   |   | Acc       |   |   | R@0.3 |   R@0.5   | R@0.7 |
> | Only RoIAlign      |  79.78 |  69.76 | 63.56 |   |   | 98.43     |   |   | 98.51 |   96.46   | 89.51 |
> | Our region encoder |  **80.68** |  **70.16** | **65.43** |   |   | **98.58**     |   |   | **98.56** |   **96. 62**  | **90.40** |
>
> ### **Q3: Evaluation of reasoning quality.**
>
> Table 4 in the manuscript presents the referring and grounding performance of AOR. Specifically:
>
> - Referring evaluates the model's ability to correctly identify anatomical regions given visual prompts.
> - Grounding measures its localization precision during reasoning.
>
> AOR attains a referring accuracy of 98.58% and a grounding R@0.7 of 90.40%, highlighting its strong capability in **visual-text alignment**.
>
> To further assess the quality of **text-level reasoning**, we randomly sample 100 test cases and evaluate the outputs by both GPT-4o and a radiologist. Each instance (ground truth and prediction) is assessed on the following dimensions (scored 1–5):
>
> - **Accuracy**: Correct identification of attributes or categories.
> - **Completeness**: Consideration of all attributes and categories relevant to the given question.
> - **Consistency**: Logical coherence and alignment between intermediate steps and final conclusions.
>
> As shown in Table 4, AOR consistently outperforms **LLaVA-o1** and **MedVLM-R1**, validating the **factuality and coherence** of its reasoning process. While absolute scores differ slightly between GPT-4o and the radiologist, the evaluation **trends are consistent**, supporting the feasibility of using LLMs as scalable proxies for expert judgment. This comparative evaluation will be added to the revised manuscript.
>
> Table 4. The score of AOR's reasoning steps.
> | Judger      | Model     | Accuracy | Completeness | Consistency |
> |-------------|-----------|:--------:|:------------:|:-----------:|
> | GPT-4o      | LLaVA-o1  |   2.52   |     3.03     |     3.86    |
> |             | MedVLM-R1 |   2.31   |     2.59     |     3.74    |
> |             | AOR(Ours) |   **3.65**   |     **4.13**     |     **4.20**    |
> | Radiologist | LLaVA-o1  |   2.68   |     2.97     |     4.11    |
> |             | MedVLM-R1 |   2.27   |     2.76     |     3.82    |
> |             | AOR(Ours) |   **3.83**   |    **4.49**     |     **4.55**    |
>
> ### **Limitations: Generalization across modalities and anatomical structures**
> **Different imaging modalities**: Taking 3D chest CT as an example, we curate a chest CT dataset under the guidance of radiologists, defining **130+ anatomical regions**. 2,070 CT scans are processed and split (80% training / 20% test). We adapt our model by replacing the 2D image encoder with a 3D variant, and evaluate its performance on report generation.
>
> As shown in Table 5, even with limited training data, our method outperforms baselines without anatomical decomposition by an average of 8.66%, demonstrating the potential generalizability of our anatomy-centric reasoning strategy to 3D modalities.
>
> Table 5. Performance  on CT region report generation.
> | Condition                    | R-L   | BERTScore | F1CheXbert |
> |------------------------------|:-------:|:-----------:|:------------:|
> | w/o anatomical decomposition | 8.09  | 68.33     | 13.94      |
> | w anatomical decomposition   | **16.45** | **79.18**     | **20.72**     |
>
> **Different tissues and Anatomical structures**: Through collaboration with radiologists, we find that the AOR paradigm can generalize to other anatomical structures as follows:
> - For organs with clearly-defined anatomical regions (e.g., brain lobes in MRI), reasoning flows can be designed around established anatomical ontologies (e.g., frontal, parietal, temporal, occipital lobes).
> - For  structures without conventional anatomical zoning, such as oral structures, reasoning can still follow structural decomposition, using components like vertebrae, ribs, crowns, roots, jaws, condyles, and airways as spatial anchors.
>
> These findings suggest that the proposed data generation pipeline and AOR architecture are adaptable to a **broad range of tissues and imaging modalities**.

---

### Official Review · Reviewer_3vmJ · 2025-07-02

**Clarity:** 3
**Significance:** 3
**Originality:** 2
**Rating:** 4
**Confidence:** 3

**Summary:**

This paper focuses on enhancing the interpretability and accuracy of medical large multimodal models (MLMMs) in chest X-ray (CXR) interpretation by introducing an anatomy-centric reasoning framework named AOR (Anatomical Ontology-guided Reasoning). The authors identify two key limitations of existing MLMMs (i.e., insufficient region-level understanding and the lack of multi-step reasoning) and address them via a three-stage training framework that incorporates anatomical ontology and Chain-of-Thought (CoT) prompting. To support this, they construct a large-scale instruction dataset, AOR-Instruction, including both visual question answering (AOR-VQA) and report generation (AOR-RG) tasks with fine-grained region–sentence alignments. Experiments across multiple benchmarks demonstrate that AOR achieves promising performance on both VQA and report generation tasks.

**Questions:**

Please refer to the weaknesses above.

**Ethical Concerns:**

["NO or VERY MINOR ethics concerns only"]

**Final Justification:**

The rebuttal has addressed my concerns, and I will be keeping my score.

**Limitations:**

Yes

**Quality:**

3

**Strengths And Weaknesses:**

Strengths
1. The proposed AOR framework integrates anatomical ontologies and region-level reasoning into medical multimodal models. This design closely mimics the diagnostic process of radiologists and enhances both interpretability and clinical relevance.
2. The authors construct a large-scale and clinically grounded instruction dataset (AOR-Instruction), including both visual question answering (AOR-VQA) and region-level report generation (AOR-RG) components. The dataset is guided by experts and offers fine-grained region–sentence alignment, making it a valuable resource for training and benchmarking future medical MLMMs.

Weaknesses
1. The paper lacks descriptions for the loss functions used in the three training stages, making it difficult to evaluate how each stage is optimized or how learning objectives are balanced.
2. From the provided code, it appears that the three-stage training is conducted separately, which may lead to error accumulation across stages. Since each stage is trained in isolation, suboptimal performance in earlier stages (e.g., incorrect region recognition) can propagate to downstream tasks like grounding and reasoning, potentially affecting overall performance and robustness. It remains unclear how the proposed framework mitigates this issue or ensures stability across stages.
3. In Table 1, AOR(Ours)-t refers to the model trained with text-only prompts, while AOR(Ours)-r/t uses both region (visual) and text prompts. However, it is unclear why in some cases the performance of AOR(Ours)-r/t is worse than that of AOR(Ours)-t. Intuitively, incorporating region-level visual information should provide additional cues for reasoning and improve performance. The paper does not provide an explanation for this performance drop.
4. In Table 3, the performance differences among the variants (especially between ID 2, 3, and 4) are relatively small and inconsistent across question types. Given these marginal gains, it is difficult to assess the true effectiveness of each representation strategy. Providing additional results on other datasets or tasks would help better demonstrate the advantages of the proposed CoT design.
5. In Table 5, Stages 1 and 2 are designed to improve region recognition and grounding, respectively. However, their impact on the downstream referring and grounding performance appears limited (i.e., adding these stages leads to only marginal gains) and in some cases, even slight drops (R@0.3: 98.48 -> 98.31). This raises questions about the effectiveness of the pretraining stages and whether their benefits consistently transfer to the final reasoning tasks. Further analysis or ablation would help clarify their contribution.
6. It would be helpful to clarify whether the proposed dataset will be publicly released, as it is also claimed as one of the key contributions of the paper.

---

> ### Author Rebuttal · Authors · 2025-07-31
>
> Thank you for your valuable comments. Below are our responses to your specific points and suggestions.
>
> ### **W1 & W2 & W5: More Details and ablation studies on our training strategy.**
>
> **Loss functions**: All three training stages use cross-entropy loss for auto-regressive language modeling.
>
> - Stage 1: The model is given an image and region features as input and trained to generate the corresponding anatomical [region name] (Loss calculation).
>
> - Stage 2 - Task 1: Given an image, region coordinates, and region features, the model is trained to predict the [region name] (Loss calculation).
>
> - Stage 2 - Task 2: The model is provided with an image and a region name as input and is trained to generate the corresponding [bounding box coordinates] (Loss calculation).
>
> - Stage 3: The model is fine-tuned on instruction-based tasks using the AOR-Instruction dataset, with outputs such as [CoT answers or reports] (Loss calculation).
>
> We will include a more detailed description of these loss functions in the revised manuscript and refer readers to Appendix B.1 for further clarification.
>
> **Three-stage training paradigm and knowledge retention**: Our training framework follows a progressive learning strategy, similar to the mainstream training paradigms of current LMMs involving referring or grounding (e.g., GPT4RoI). Specifically:
>
> - Stage 1 improves anatomical region recognition.
> - Stage 2 jointly optimizes anatomical region recognition and localization.
> - Stage 3 benefits from the prior stages and further enhances the model's region recognition and localization capabilities while optimizing reasoning.
>
> Although the stages are not jointly optimized, knowledge is preserved via model checkpoints and parameter initialization. This mitigates error accumulation, as the pretrained representations are progressively refined rather than discarded or overwritten.
>
> **Effectiveness of Our Training Strategy**:  As shown in Table 5 of the manuscript, the inclusion of Stage 1 and Stage 2 leads to an average improvement of 2.15% on the VQA task.
>
> To further validate the effectiveness of our training strategy, we conduct an ablation study on the VQA-RAD dataset under the zero-shot setting. As shown in Table 1, VQA accuracy increased by 3.74%, while referring and grounding accuracies improved by an average of 5.46% and 4.31%, respectively. The inclusion of Stage 1 and Stage 2 enhances the model’s stability and robustness and yields relatively more significant gains on the out-of-distribution (OOD) test set.
>
>
> **Further analysis of the Referring and Grounding results can be conducted from the following two perspectives**:
> - Compared to MIMIC-CXR-VQA, both Referring and Grounding performance metrics decline on the VQA-RAD dataset. This is primarily because VQA-RAD serves as an out-of-distribution (OOD) dataset, where some questions involve anatomical regions not included in our predefined set of 29 zones. These regions, such as the aortopulmonary window (an anatomical gap between the aortic arch and pulmonary artery), are rarely encountered in routine clinical practice and are more clearly visualized in chest CT or contrast-enhanced imaging, rather than on standard chest X-rays, where they are not a primary focus.
>
> - On VQA-RAD, the inclusion of Stage 1 and Stage 2 training leads to more noticeable gains. This can be attributed to the fact that the training data used in Stage 3 is skewed toward anatomically frequent and diagnostically dominant regions in chest X-rays, such as the lungs and heart. In contrast, VQA-RAD often includes regions that appeared less frequently in our Stage 3 training data, e.g.,  the trachea. According to our statistics from a collaborating hospital, the incidence of tracheal abnormalities in chest X-rays from 2018 to 2022 was less than 0.01%. In Stage 1 and Stage 2, however, the distribution of anatomical regions is balanced, allowing the model to learn to better recognize and localize underrepresented regions.
>
> Table 1. Effectiveness of our training strategy on VQA-RAD (zero-shot setting).
> | **Strategy** |             | **VQA-RAD** |           |   |   | **Referring** |   |   |           | **Grounding** |           |
> |:--------------:|:-------------:|-------------|-----------|---|---|---------------|---|---|:---------:|:-------------:|:---------:|
> | **Stage 1**  | **Stage 2** | **closed**  | **open**  |   |   | **Acc**       |   |   | **R@0.3** |   **R@0.5**   | **R@0.7** |
> | **✗**        | **✗**       | 54.00       | 21.04     |   |   | 87.27         |   |   |   82.14   |     76.79     |   69.64   |
> | **✓**        | **✗**       | 56.16       | 22.53     |   |   | 89.09         |   |   |   84.21   |     78.18     |   70.90   |
> | **✓**        | **✓**       | **57.53**   | **24.99** |   |   | **92.73**     |   |   | **86.79** |   **81.13**   | **73.58** |
>
> These results confirm that the early-stage training benefits generalization and performance in later reasoning tasks. We will revise the manuscript to clarify these findings and their implications.
>
> ### **W3: Performance Comparison Between AOR-t and AOR-r/t**.
>
> Thank you for pointing out this. The setting of AOR(Ours)-r/t is designed to demonstrate our model’s flexibility in handling **multimodal input formats** during inference, specifically, both textual and visual prompts (e.g., bounding boxes drawn directly on the image). While this reflects a more general and practical usage scenario, it also introduces additional complexity, as the model must dynamically parse and reason over **heterogeneous input modalities**. The slight performance drop observed in AOR-r/t is not indicative of a failure but rather a result of the increased input diversity and interpretive burden placed on the model. We will clarify this distinction and the implications in the revised manuscript.
>
> In addition, it’s important to note that both AOR-t and AOR-r/t utilize region-level visual information to enhance reasoning, consistently reasoning with regions throughout the whole process. For more details, please refer to Figure C6 in Appendix.
>
>
> ### **W4: Analysis and Additional Results on Different CoT Representations**.
>
> **Inconsistency Across Question Types**: The observed performance "inconsistency" is primarily due to differences in **question difficulty** across types. As described in Section 5.1 of the manuscript:
>
> - **Verify-type questions** require only a binary yes/no response. Even without fully understanding the question, the model has a 50% chance of answering correctly by guessing.
> - **Choose-type questions** are evaluated strictly: selecting one incorrect option or omitting a correct one results in the entire answer being marked incorrect. These questions require the model to reason about multiple lesions and anatomical regions.
> - **Query-type questions** also demand precise generation of comprehensive answers. A correct response necessitates full and accurate understanding of the image and question.
>
> Therefore, the benefits of coherent, logically structured reasoning are more evident in the more complex **choose** and **query** tasks.
>
> **Performance Across Model Variants (IDs)**: We elaborate from the following three perspectives:
>
> - **Further analysis of Table 3 in the manuscript**: **ID1 → ID2 (Effect of CoT)**: As shown in Table 3 of the manuscript, incorporating our CoT strategy significantly improves performance across all question types. This early gain establishes a strong baseline, making subsequent improvements more challenging. **ID2 → ID4 (Effect of Spatial and Visual Cues)**: Despite the strong baseline, introducing spatial and visual cues in later stages still yields additional performance gains, with a 2.64% improvement on choose questions and a 2.03% improvement on query questions. This highlights the added value of spatial and visual information to fine-grained reasoning tasks.
>
> - **Zero-shot Evaluation on VQA-RAD**: We conduct an additional ablation on the VQA-RAD dataset under the zero-shot setting. As shown in Table 2, the inclusion of spatial and visual cues improves accuracy by 5.18% on closed-type questions and 5.79% on open-type questions. These gains further confirm the model’s enhanced reasoning and generalization under unseen distributions.
>
> - **Interpretability and Practical Value**: Finally, we emphasize that our final model (ID-4) not only achieves the highest answer accuracy, but also offers enhanced interpretability and practical value by visualizing the approximate lesion regions attended to during reasoning, providing valuable insights in clinical scenarios.
>
> Table 2. Comparison of different CoT representations on VQA-RAD.
> | Exp_ID |  coor | region |  CoT  |   closed  |    open   |
> |:------:|:-----:|:------:|:-----:|:---------:|:---------:|
> |    1   | **✗** |  **✗** | **✗** |   47.74   |   16.90   |
> |    2   | **✗** |  **✗** | **✓** |   52.05   |   19.20   |
> |    3   | **✓** |  **✗** | **✓** |   54.50   |   21.91   |
> |    4   | **✓** |  **✓** | **✓** | **57.53** | **24.99** |
>
> ### **W6: Dataset Release Plan.**
> We confirm that the proposed dataset **will be publicly released**. We are currently organizing the files and documentation and will upload the dataset to **PhysioNet** as soon as possible.

---

> > ### Author Response · Authors · 2025-08-07
> > **Sincerely looking forward to further discussion**
> >
> > Dear Reviewer 3vmJ,
> >
> > We deeply appreciate the time and effort you have dedicated to reviewing our paper.
> >
> > As the discussion period is approaching its end, we would like to inquire if our response addressed your concerns.
> >
> > Please let us know if you have any further questions or feedback.

---

> > > ### Comment · Reviewer_3vmJ · 2025-08-08
> > >
> > > Thank you for your detailed response. It has addressed my concerns, and I will be keeping my score.

---

### Official Review · Reviewer_CaAU · 2025-07-06

**Clarity:** 2
**Significance:** 2
**Originality:** 3
**Rating:** 4
**Confidence:** 3

**Summary:**

To tackle the problem of insufficient region-level understanding and interaction, as well as limited accuracy and interpretability due to single-step prediction in MLMMs for chest X-ray interpretation, this paper proposes an AOR framework. The AOR framework enhances MLMMs with anatomy-centric reasoning capabilities, enabling multimodal, multi-step reasoning through region-level information. Additionally, the paper develops the AOR-Instruction dataset, which includes anatomical regions and their ontologies, to support training MLMMs for improved accuracy and interpretability in  VQA and report generation tasks.

**Questions:**

Please carefully address Cons 1. My evaluation score would drop if more recent methods is not included and compared.

**Ethical Concerns:**

["NO or VERY MINOR ethics concerns only"]

**Final Justification:**

My concerns have been resoloved and I will keep my score.

**Limitations:**

yes

**Paper Formatting Concerns:**

Not noticed.

**Quality:**

3

**Strengths And Weaknesses:**

Pros:
1.The paper introduces a novel AOR framework that enhances MLMMs with anatomy-centric reasoning for improved medical imaging analysis.
2.The methodology and experimental setup are clearly described. Illustrations in supplementary are well put.
3.The authors validate their approach using extensive datasets, demonstrating the effectiveness and robustness of the AOR framework.

Cons:
1.My main concern is the baselines are not up-to-date. The work below, published in 1st April this year, proposed LLaVa-Rad that brings dramatic improvements over LLaVa-Med. The reported performances may surpass the current work. Though the current method has merits of its own, it’s more complete to include LLaVa-Rad in baselines. (Zambrano Chaves J M, Huang S C, Xu Y, et al. A clinically accessible small multimodal radiology model and evaluation metric for chest X-ray findings[J]. Nature Communications, 2025, 16(1): 3108.)

Minors:
1.Line 6, 7: “Insufficient” and "Limited" should not be capitalized?
2.Line 35: A comma is needed before "but" to separate the clauses.

---

> ### Author Rebuttal · Authors · 2025-07-31
>
> We are grateful for your valuable feedback and suggestions. We hope that the following responses address your concerns.
>
>
> ### **W1: Include more up-to-date baselines.**
>
> In response to the reviewer’s concern, we have additionally compared three representative up-to-date baselines on VQA, full image report generation, and region report generation tasks:
>
> - **Qwen2.5-VL** [1], a recent general-domain large multimodal model,
> - **MedRegA** [2], a region-aware bilingual medical large multimodal model, and
> - **LLaVA-Rad** [3], a model specifically designed for radiology report generation.
>
> **VQA Performance**: As shown in Table 1, our model outperforms Qwen2.5-VL by an average of 8.06% and MedRegA by 4.50% on the VQA task. These improvements demonstrate the strength of our carefully designed multimodal reasoning framework, particularly in lesion recognition and localization.
>
> > ***Note***:
> > - LLaVA-Rad was exclusively trained for the report generation task and lacks VQA capabilities; therefore, it is not included in this comparison.
> > - Due to the large size of MedRegA (40B), we are currently unable to provide its fine-tuned results on the MIMIC-CXR-VQA dataset within a short timeframe. We will include these results in the revised version.
>
> Table 1. Performance on VQA. Results in *italics* indicate that the training data of MedRegA includes VQA-RAD.
> | Method     | Res | Params |           | MIMIC-CXR-VQA |       |   |   | VQA-RAD |           |   |   | CheXpert |       |
> |------------|:---:|:------:|:-----------:|:-------------:|:-------:|---|---|---------|-----------|---|---|----------|-------|
> |            |     |        | verify    |     choose    | query |   |   | closed  | open      |   |   | closed   | open  |
> | Qwen2.5-VL | 336 |   7B   | 75.18     |     56.66     | 61.46 |   |   | 61.64   | 23.22     |   |   | 61.78    | 37.20 |
> | MedRegA    | 448 |   40B  | -         |       -       | -     |   |   | ***66.46***   | *27.82*     |   |   | 71.49    | 32.85 |
> | Ours       | 336 |   7B   | **80.48** |   **71.96**   | **65.05** |   |   | 63.01   | **28.19** |   |   | **71.58**    | **53.85** |
>
> **Full Image Report Generation**: As shown in Table 2, our method achieves an average performance gain of **12.18%** over Qwen2.5-VL, **4.47%** over MedRegA, and **0.58%** over LLaVA-Rad, when evaluating fairly at the same input resolution of 336. The performance of our model in R-L and BERTScore even comes close to that of the LLaVA-Rad model using a 518 resolution, and increasing the resolution of our model would be a promising future direction.
>
> >***Note***: The reported metrics differ slightly from those in the original LLaVA-Rad paper, mainly due to two reasons: (1) LLaVA-Rad evaluates on the full test split of MIMIC-CXR, containing 2461 image-report pairs, whereas our evaluation is based on 500 image-report pairs from the MIMIC-CXR-VQA test set which is a gold standard dataset manually validated and corrected by clinicians; (2) LLaVA-Rad focuses on generating the "findings" section, while our ground truth includes both the "findings" and the "impression" sections, the latter being an  essential component of a complete radiology report. These factors contribute to the observed discrepancy in reported performance.
>
> Table 2. Performance on full image report generation.
> | Method      | Res | Params |  R-L  | BERTScore | F1CheXbert |
> |-------------|:---:|:------:|:-----:|:--------:|:----------:|
> | Qwen2.5-VL | 336 |   7B   |  12.80 |   76.34  |    30.15   |
> | MedRegA     | 448 |   40B  | 21.78 |   83.29  |    37.33   |
> | LLaVA-Rad   | 518 |   7B   | **25.68** |   **84.94**  |    **50.52**   |
> | LLaVA-Rad   | 336 |   7B   | 24.22 |   83.67  |    46.20   |
> | Ours        | 336 |   7B   | 25.37 |   83.92  |    46.53   |
>
> **Region Report Generation**: As shown in Table 3, our model demonstrates strong interpretability and fine-grained capability in generating region-specific descriptions in response to user-provided textual and visual prompts. Compared to existing  baselines, it achieves substantial improvements in this task.
>
> Table 3. Performance on region report generation.
> | Method      | Res | Params |    R-L    |  BERTScore | F1CheXbert |
> |-------------|:---:|:------:|:---------:|:---------:|:----------:|
> | Qwen2.5-VL | 336 |   7B   |   11.07   |   78.81   |    12.26   |
> | MedRegA     | 448 |   40B  |   16.93   |   80.85   |    26.33   |
> | LLaVA-Rad   | 518 |   7B   |   18.77   |   81.90   |    33.51   |
> | LLaVA-Rad   | 336 |   7B   |   18.71   |   81.80   |    31.40   |
> | Ours        | 336 |   7B   | **35.11** | **84.54** |  **36.65** |
>
> These up-to-date baselines will be incorporated into the revised manuscript to strengthen the evaluation.
>
> We appreciate your attention to detail. “Insufficient” and “limited” will be decapitalized as suggested. We will also add the appropriate comma before “but” on Line 35.
>
> ---
>
> [1] Bai, S., Chen, K., Liu, X., et al. *Qwen2.5-VL Technical Report*. arXiv preprint, 2025.
>
> [2] Wang L, Wang H, Yang H, et al. *Interpretable bilingual multimodal large language model for diverse biomedical tasks.* ICLR 2025.
>
> [3] Chaves, J. M. Z., Huang, S. C., Xu, Y., et al. *Towards a Clinically Accessible Radiology Foundation Model: Open-Access and Lightweight, with Automated Evaluation*. Nature Communications, 2025.

---

> > ### Author Response · Authors · 2025-08-07
> > **Sincerely looking forward to further discussion**
> >
> > Dear Reviewer CaAU,
> >
> > We sincerely thank you for your time and insightful comments on our submission.
> >
> > As we are nearing the end of the discussion period, we would be grateful if you could let us know if our response addressed your concerns or if there are any further questions or feedback.

---

> > > ### Comment · Reviewer_CaAU · 2025-08-08
> > >
> > > Thanks for providing further evidences. My concerns have been resoloved and I will keep my score.

---

### Note · Authors · 2025-08-15

We sincerely thank the ACs and reviewers for their time and valuable feedback. We are encouraged by the recognition of several key strengths:

- **AOR-Instruction Dataset:** We developed a large instruction dataset guided by expert physicians, using step-by-step reasoning over anatomical regions. Reviewers described the pipeline as “interpretable, sound and effective” (SRND), and considered the dataset “a valuable resource for training and benchmarking future medical MLMMs” (3vmJ) that “could benefit the broader medical AI community” (wKs5).
- **AOR Framework:** We proposed an Anatomical Ontology-Guided Reasoning (AOR) framework that supports both textual and optional visual prompts, empowering Medical LMMs with “anatomy-centric reasoning capabilities” (CaAU). This design “closely mimics the diagnostic process of radiologists and enhances both interpretability and clinical relevance” (3vmJ), which is “especially valuable in high-stakes domains like healthcare” (wKs5).
- **Extensive Experiments:** Extensive experiments demonstrated AOR’s effectiveness across both VQA and report generation tasks (CaAU, SRND, wKs5).

In response to reviewers’ suggestions, we made additional enhancements and are pleased that **all reviewers agreed their concerns have been addressed**:
- **More Details:** We added implementation details on the region encoder (SRND), training loss and model configurations (3vmJ), and quality control processes (wKs5).
- **Additional Experimental Validation:**
  - **Broader Baseline Comparisons:** We included recent models Qwen2.5-VL (Tech Report, Feb 2025), MedRegA (ICLR 2025), and LLaVA-Rad (Nature Communications 2025). AOR outperforms both larger (MedRegA, 40B) and more specialized (LLaVA-Rad) models in VQA and report generation, demonstrating its strong anatomy-centric reasoning (CaAU).
  - **Additional Ablations:** We conducted ablation studies on the OOD VQA-RAD dataset, further verifying the contribution of each module (3vmJ, SRND, wKs5).
  - **Reasoning Quality:** Beyond Table 5 of the manuscript, we added human and LLM-as-a-Judge (GPT-4o) evaluations of AOR’s reasoning on a 100-question subset, further confirming its soundness (SRND).
- **More Discussion:** We added experimental results on 3D chest CT and discussed extensions to other modalities (e.g., dental imaging), affirming AOR’s potential for generalization (SRND, wKs5).

We again thank ACs and reviewers and hope our work contributes to advancing robust and interpretable medical AI.

---

### Decision · Program_Chairs · 2025-09-17

**Decision:**

Accept (poster)

**Comment:**

The paper presents AOR (Anatomical Ontology-Guided Reasoning), a new framework that enhances the reasoning capabilities of large multimodal models when interpreting chest X-rays. AOR mimics how radiologists approach diagnoses by reasoning anatomically, and it is supported by a new dataset, AOR-Instruction, crafted from expert input to promote step-by-step clinical reasoning. The approach aims to improve accuracy, interpretability, and generalizability across multiple tasks, including visual question answering and report generation.

Initial reviewer concerns focused on missing comparisons to recent baselines like LLaVA-Rad, limited training detail, and uncertainty about the added value of some model components. The authors responded thoroughly, introducing new experiments, results, and analyses to show AOR’s improvements across a wide range of settings. These included new baselines, validation on 3D CT data, and a detailed component analysis supported by human and GPT-4o evaluations.

After the rebuttal, all reviewers affirmed their positive scores and noted that their concerns were resolved. The final ratings are one Accept, and three Borderline Accepts. Considering the paper's positive scores and contributions to interpretable and robust AI in medical imaging, the area chair recommends Accept (Poster).